# The insect somatostatin pathway gates vitellogenesis progression during reproductive maturation and the post-mating response

Chen Zhang[1], Anmo J. Kim[2,3], Crisalesandra Rivera-Perez[4], Fernando G. Noriega[5,6] & Young-Joon Kim [1✉]

Vitellogenesis (yolk accumulation) begins upon eclosion and continues through the process of sexual maturation. Upon reaching sexual maturity, vitellogenesis is placed on hold until it is induced again by mating. However, the mechanisms that gate vitellogenesis in response to developmental and reproductive signals remain unclear. Here, we have identified the neuropeptide allatostatin-C (AstC)-producing neurons that gate both the initiation of vitellogenesis that occurs post-eclosion and its re-initiation post-mating. During sexual maturation, the AstC neurons receive excitatory inputs from Sex Peptide Abdominal Ganglion (SAG) neurons. In mature virgin females, high sustained activity of SAG neurons shuts off vitellogenesis via continuous activation of the AstC neurons. Upon mating, however, Sex Peptide inhibits SAG neurons, leading to deactivation of the AstC neurons. As a result, this permits both JH biosynthesis and the progression of vitellogenesis in mated females. Our work has uncovered a central neural circuit that gates the progression of oogenesis.

[1] School of Life Sciences, Gwangju Institute of Science and Technology (GIST), 123 Cheomdangwagi-ro, Buk-gu, Gwangju 61005, Republic of Korea. [2] Department of Electronic Engineering, Hanyang University, 222 Wangsimni-ro, Seongdong-gu, Seoul 04763, Republic of Korea. [3] Department of Artificial Intelligence, Hanyang University, 222 Wangsimni-ro, Seongdong-gu, Seoul 04763, Republic of Korea. [4] Department of Fisheries Ecology, Consejo Nacional de Ciencia y Tecnología, Centro de Investigaciones Biológicas del Noroeste, 23096 La Paz, Baja California Sur, Mexico. [5] Department of Biological Sciences and Biomolecular Science Institute, Florida International University, Miami, FL, USA. [6] Department of Parasitology, University of South Bohemia, České Budějovice, Czech Republic. ✉email: kimyj@gist.ac.kr

In *Drosophila melanogaster*, ovary morphogenesis begins in the third larval instar and is completed within 48 hours after pupariation[1]. The ovary contains about 16 ovarioles, each of which represents an independent egg assembly line with progressively developing follicles (egg chambers). This assembly line begins with dividing germline stem cells (GSC) at one end and ends with mature eggs at the other. During oogenesis, a GSC divides asymmetrically to produce a daughter GSC and a cystoblast[2]. The cystoblast subsequently divides four times to produce a cystocyte complex comprising one oocyte and 15 nurse cells. A layer of follicular epithelial cells adheres to and surrounds the oocyte-nurse cell complex to produce a stage 1 follicle. During maturation, follicles undergo sequential transitions from pre-vitellogenesis (stages 1–7) to vitellogenesis (i.e., yolk accumulation) (stages 8–14)[3]. In *D. melanogaster*, females molt into the adult stage (i.e., undergo eclosion) with ovaries that lack vitellogenic follicles[2]. Vitellogenesis begins after eclosion and continues throughout reproductive maturation.

Vitellogenesis initiation is an important control point in oogenesis that integrates hormonal cues to match female physiological conditions (i.e., nutrition, mating status, etc.). The major gonadotropic hormones in *D. melanogaster* are juvenile hormone (JH) and the insect steroid hormones referred to as ecdysteroids[3]. Ecdysteroids stimulate yolk protein (YP) synthesis in the fat body[4], while JH stimulates the synthesis and uptake of YP by the ovary[4,5]. Thus, JH is essential for the continuing development of vitellogenic follicles past stages 8 and 9. As a female fly completes reproductive maturation, JH levels fall gradually and vitellogenesis ceases. Later, mating once more stimulates the oogenesis that is required to sustain robust egg-laying activity in mated females. In *D. melanogaster*, a male-derived seminal fluid protein called sex peptide (SP) elicits oogenesis progression[6]. SP induces GSC proliferation by stimulating the biosynthesis of ecdysteroids[7]. The mechanism by which SP regulates JH biosynthesis and vitellogenesis, however, remains unclear.

The sperm-containing seminal fluid males transfer to females during copulation is composed of a wide range of chemical substances[8]. Not only does seminal fluid provide a supportive milieu for sperm survival, it also modifies the physiology and behavior of recipient females to maximize sperm fertility. SP is the most well-studied seminal fluid substance in *D. melanogaster*. SP is a 36-mer peptide produced by the male accessory gland[9] and attached to sperm tails. SP is transferred to females and stored in the female sperm storage organs. Because it is released continuously from sperm by proteolysis of a trypsin-like cleavage site, it can sustain the post-mating state as long as sperm remain in the sperm storage organs (typically about a week)[10]. SP acts through SP receptor (SPR)—a G-protein coupled receptor that functions via Gαi or Gαo—to silence SPR-expressing peripheral sensory neurons (SPSNs)[11–14]. SPSNs innervate the lumen of the uterus and send axonal processes into the tip of the abdominal ganglion, relaying the SP signal to Mip-vAL and SAG neurons. Mip-vAL neurons connect SPSNs and SAG neurons in the abdominal ganglion[15]. SAG neurons project to the dorsal protocerebrum in the brain and regulate SP-associated behaviors and physiological responses. These include the suppression of mating receptivity[16], vaginal plate opening[17], stimulation of oviposition[18], increasing salt preference[19], and reduced siesta sleep[20]. Thus, the SP signal seems to diverge after passing through the SAG neurons to regulate post-mating behaviors and physiological changes. The recent discovery of the neural pathways that implicate SAG neurons in oviposition and mating receptivity (i.e., vaginal plate opening) supports this notion[17,18]. Still, the circuits downstream of the SAG neurons that regulate vitellogenesis remain unknown.

In this study, we identified two pairs of allatostatin C (AstC)-producing thoracic ganglion neurons (*AstC-mTh*) that link SAG neurons with JH biosynthesis and vitellogenesis initiation, not only in mated females, but also in young virgin females during reproductive maturation. AstC was first identified in the hawk-moth *Manduca sexta* based on its "allatostatic" activity (i.e., inhibition of JH biosynthesis) against the JH-producing endocrine organ referred to as the *corpora allata* (CA)[21]. An allatostatic function was also ascribed to AstC in *D. melanogaster* when Wang et al. (2012) found that knockdown of either AstC or AstC receptors increases JH-III levels[22]. In this study, we found that SAG neurons function upstream of *AstC-mTh* neurons, which secrete AstC and inhibit JH biosynthesis in the CA via AstC receptors (star1 and AstC-R2). As a young virgin female completes reproductive maturation, SAG neuron activity rises, augmenting the AstC-induced inhibition of the CA. In mated females, the SP signal attenuates SAG neuron activity and, in turn, the secretion of AstC from *AstC-mTh* neurons. This reduces AstC-induced inhibition of the CA, eventually permitting JH biosynthesis and vitellogenic oocyte development.

## Results

**AstC gates vitellogenesis during reproductive maturation.** In *D. melanogaster*, adult females emerge with ovaries lacking vitellogenic follicles (i.e., follicles older than stage 8). To better understand the progression of vitellogenesis that takes place during reproductive maturation, we examined the ovaries of virgin females, counting early (stages 8–11) and late vitellogenic follicles (stages 12–14) separately over the course of 5 days after eclosion. We found control ovaries contained vitellogenic follicles as early as 12 hours post-eclosion (yellow circle in Fig. 1a). Early vitellogenic follicles accumulated quickly and reached maximum numbers ($17 \pm 3$ oocytes per female) within 24 h. They then remained elevated for an additional 24 h before declining again and reaching a lower basal level thereafter. Later-stage vitellogenic follicles (stages 12–14) began to appear 24 h after eclosion. They then continued to increase over 3–4 days before reaching a maximum. Thus, vitellogenesis during reproductive maturation seems to progress in two distinct phases. The initial phase, characterized by a rapid accumulation of early vitellogenic follicles, begins soon after eclosion and lasts 48 h. In the subsequent phase, vitellogenesis initiation slows down until coming to an end 72 h after eclosion. This maintains the number of early vitellogenic follicles at roughly 31–55% of its peak.

Considering the role of AstC in generating the circadian vitellogenesis rhythm[23], we examined post-eclosion vitellogenesis in virgin females lacking a functional *AstC* allele. As with the wild-type controls described above, the ovaries of AstC-deficient (*AstC[1]/AstC[1]*) females contained no vitellogenic follicles when examined immediately after eclosion. Twelve hours later, however, AstC-deficient ovaries produced considerably more early-stage follicles ($10 \pm 1$ follicles per female) than controls ($3 \pm 1$ follicles per female). This difference between the two groups reached statistical significance at both 12 and 24 h after eclosion, but not at 48 h after eclosion or thereafter (Fig. 1a and S1a). Thus, AstC seems to delay the initial phase of vitellogenesis post-eclosion. The precocious onset of vitellogenesis caused by AstC deficiency did not stimulate egg-laying activity in the virgin females (Fig. S1b). It also did not seem to compromise viability of eggs or progeny, because AstC deficiency has a limited impact on the rates of egg-to-pupa and pupa-to-adult development (Fig. S1c).

Next, we restored AstC expression in *AstC-Gal4* cells and again examined vitellogenesis progression post-eclosion. As expected, we found restoration of AstC in *AstC-Gal4* cells

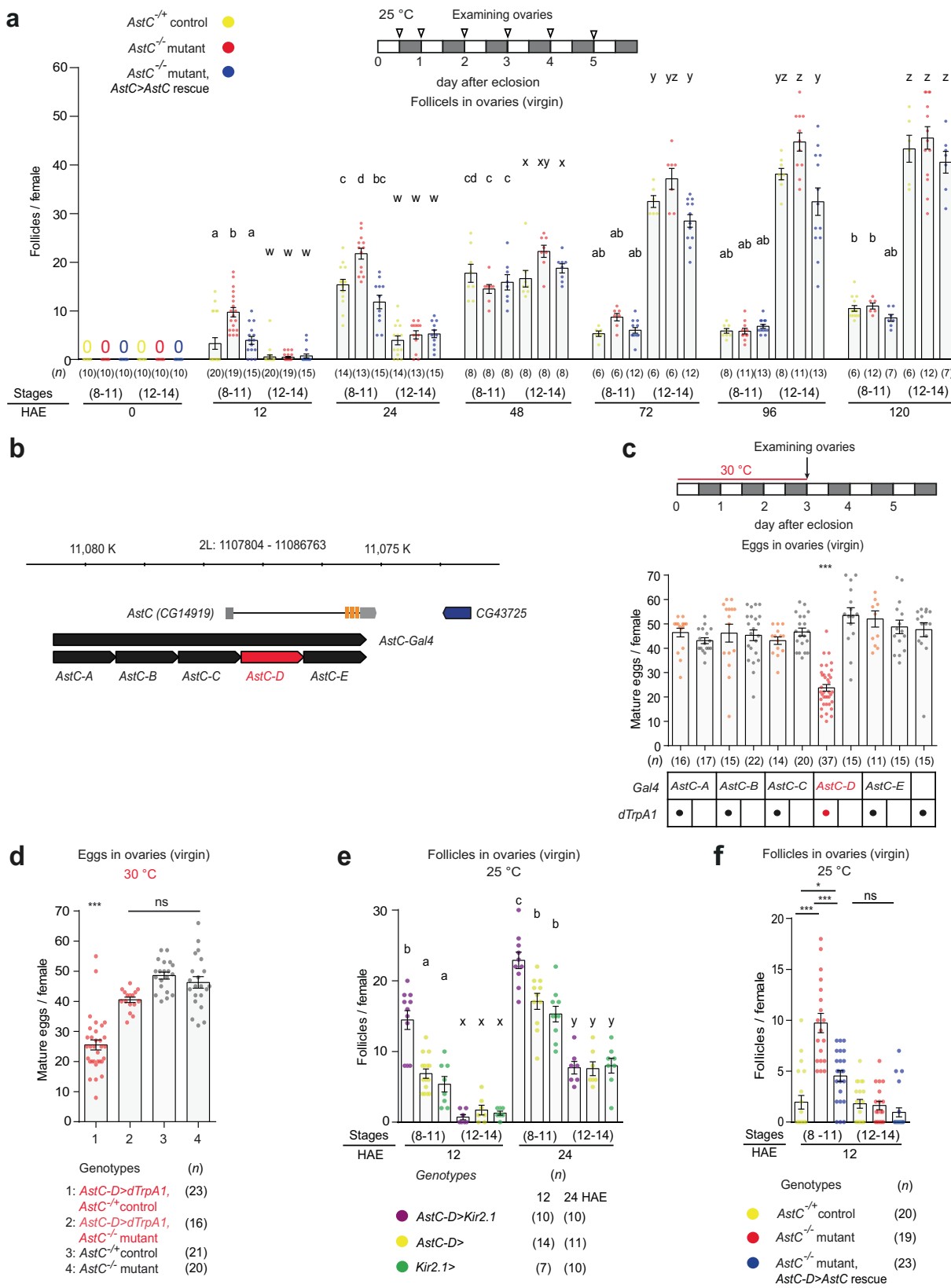

reduced the number of early- and late-stage vitellogenic oocytes to control levels (compare the yellow and blue circles in Fig. 1a). *AstC-Gal4* drives expression in many central nervous system (CNS) neurons[23]. Among these, six pairs of neurons referred to

as circadian dorsal neuron 1 (*AstC-DN1*) are involved in vitellogenesis initiation, specifically with its circadian rhythm. Unlike *AstC-Gal4*, however, restoration of AstC expression in *AstC-DN1* alone failed to restore the number of early

**Fig. 1 AstC from *AstC-D-Gal4* neurons gates vitellogenesis during reproductive maturation. a** Above, experimental protocol for **a**, **e**, and **f**. Below, number of stage 8–11 and 12–14 follicles per virgin female of the indicated genotypes at the indicated hours after eclosion (HAE). The letters above the columns in **a** and **e** indicate significant differences ($p < 0.05$) for comparisons of time points and genotypes (two-way ANOVA with Bonferroni post-hoc test for multiple comparisons); **a**–**d** for stage 8–11 and w–z for stage 12–14. Error bars in **a** and **c**–**f** indicate s.e.m. **b** AstC gene structure and genomic fragments (black or red bars) used to generate *AstC-Gal4* and the other related *Gal4*, *Gal4AD*, and *Gal4DBD* transgenes. **c**, **d** Above in **c**, experimental protocol for **c** and **d**. Number of mature eggs per virgin female 3 days after eclosion of the indicated genotypes at 30 °C. One-way ANOVA followed by Tukey's test for multiple comparisons; ***$p < 0.001$; ns (non-significance) and no labeling, $p > 0.05$. **e** Number of stage 8–11 and 12–14 follicles per virgin female of the indicated genotypes at the indicated HAE. **f** Number of stage 8–11 and 12–14 follicles per virgin female of the indicated genotypes at 12 HAE. One-way ANOVA followed by Tukey's test for multiple comparisons; ***$p < 0.001$, *$p < 0.05$, ns (non-significance), $p > 0.05$. For a summary of statistical analyses including adjusted $p$ values and a detailed list of genotypes, see Supplementary Tables 1 and 2, respectively.

vitellogenic oocytes to control levels (Fig. S1d). Thus, *AstC-DN1* is unlikely involved in the vitellogenesis associated with reproductive maturation.

**AstC neurons that function during reproductive maturation.** Consistent with an inhibitory role for the AstC neurons that arrest vitellogenesis associated with reproductive maturation, activation of the *AstC-Gal4* neurons of virgin females for three days shortly after eclosion significantly reduces total oogenesis[23]. To map the responsible AstC neurons, we restricted the expression of the *AstC-Gal4* driver to distinct subsets of neurons by preparing five additional Gal4 transgenes (*AstC-A*, *AstC-B*, *AstC-C*, *AstC-D*, and *AstC-E*), each carrying ~1-kb genomic fragments that together tile the 5′-upstream cis-regulatory region of *AstC* used to generate the original *AstC-Gal4* (Fig. 1b). We then used these additional Gal4 drivers to drive expression of the warmth-activated cation channel dTrpA1, thus permitting the specific activation of subsets of the neurons targeted by the original AstC-Gal4 transgene. Among these five new Gal4 lines, *AstC-D-Gal4* activation led to a significant reduction in total oogenesis. Females carrying both *AstC-D-Gal4* and *UAS-dTrpA1* produced ~50% fewer mature eggs at 30 °C than control females carrying *AstC-D-Gal4* alone (Fig. 1c). A delay in oogenesis progression seems to be responsible for the reduction in total oogenesis. Neither *AstC-D-Gal4* neuronal activation nor silencing stimulated egg-laying in virgin females (Fig. S1e, f).

Next, we asked whether the AstC neuropeptide is required for the reduction in total oogenesis caused by activation of *AstC-D-Gal4* neurons. Indeed, the activation of *AstC-D-Gal4* neurons resulted in a significant reduction in total oogenesis in control females ($AstC^{-/+}$), but not in AstC-deficient females ($AstC^{-/-}$) (Fig. 1d). Next, we further examined the role *AstC-D-Gal4* neurons play in vitellogenesis by using them to drive the expression of the inwardly-rectifying potassium channel Kir2.1[24] or the temperature-sensitive dynamin mutant *shibire* ($Shi^{ts}$)[25], and then counting the number of early and late vitellogenic follicles at 12 and 24 h after eclosion. We found silencing *AstC-D-Gal4* neurons produced a phenocopy of AstC deficiency, leading young virgin females to produce more early-stage vitellogenic follicles (stages 8–11) than controls (Figs. 1e and S1g). This is precisely what would be expected if these are indeed the AstC neurons of interest—negative regulators of the vitellogenesis associated with reproductive maturation. Moreover, we confirmed that restoring AstC expression only in *AstC-D-Gal4* neurons can rescue vitellogenesis in AstC-deficient females to wild-type levels (Fig. 1f). On the strength of these data, we conclude that *AstC-D-Gal4* targets a subset of AstC neurons that gates vitellogenesis-associated reproductive maturation.

**Identification of *AstC-mTh* neurons.** To further restrict *AstC-D-Gal4* activity to a smaller group of cells, we employed the split-Gal4 system[26]. Genomic DNA fragments used to generate *AstC-Gal4* and *AstC-D-Gal4* were fused with the Gal4 DNA-binding (DBD) and transcription activation domains (AD), respectively, and combined to produce *AstC-mTh-Gal4* (i.e., *AstC-Gal4DBD; AstC-D-Gal4AD*). Compared with *AstC-Gal4* and *AstC-D-Gal4*, *AstC-mTh-Gal4* showed far less brain and abdominal ganglion expression. Instead, its activity seemed restricted to several neurons in the thoracic ganglia (compare Fig. S2c with Fig. S2a, b). We found that thermal activation of *AstC-mTh-Gal4* neurons suppressed total oogenesis to a level comparable to that of *AstC-D-Gal4* neuron activation (i.e., compare Fig. 2a with Fig. 1d). Moreover, when we used *AstC-mTh-Gal4* to drive expression of *AstC-RNAi*, we observed a nearly complete de-repression of the oogenesis inhibition caused by *AstC-mTh-Gal4* neuron activation (Fig. 2a). Of note, expression of *AstC-RNAi* in *AstC-mTh-Gal4* neurons also led to a nearly complete loss of anti-AstC expression in the thoracic ganglion and dorsal subesophageal zone (SEZ) (arrows in Fig. S2d, e).

When we examined the CNS of flies expressing *UAS-mCD8-EGFP* in *AstC-mTh-Gal4* neurons, we found 40 cells with neuron-like morphology (Fig. 2b). Of these cells, 6 in the brain and 8 in the ventral nerve cord (VNC), including 4 thoracic ganglion cells, were also positive for anti-AstC (orange circles in Fig. 2c). To visualize and examine the function of *AstC-mTh-Gal4* neurons in the brain, we combined *AstC-mTh-Gal4* with the $Otd^{FLP}$ transgene[27,28], which is active exclusively in the brain, not the VNC (Fig. S2f). Unlike activation of *AstC-mTh-Gal4* neurons (including those in the VNC), activation of *AstC-mTh-Gal4* and $Otd^{FLP}$ double-positive brain neurons had a limited impact on total oogenesis (Fig. 2d). This suggests the *AstC-mTh-Gal4* neurons in the brain are unlikely responsible for oogenesis inhibition.

Since the silencing of AstC neurons (i.e., *AstC-D-Gal4* neurons) seems to expedite post-eclosion vitellogenesis by ~12 h, AstC neurons associated with reproductive maturation should already be active prior to eclosion. Thus, we monitored neural activity in *AstC-mTh-Gal4* neurons using TRIC (i.e., transcriptional reporter of intracellular $Ca^{2+}$), which increases GFP expression in proportion to $[Ca^{2+}]_i$[29]. We observed a robust TRIC signal exclusively in a pair of AstC neurons in the mesothoracic ganglion (hereafter referred to as *AstC-mTh* neurons) (arrows in Fig. 2e, S2g), which remained unchanged in all examined time points from −1 to 3 days post-eclosion (Fig. S2h). The TRIC staining we observed was so strong that it labeled the entire neuronal arbor of each *AstC-mTh* neuron. This suggests that a ceiling effect may have masked any further increase in TRIC activity associated with reproductive maturation. At low frequency (~10%), we observed TRIC labeling of only a single *AstC-mTh* neuron, revealing its anatomy at single-cell resolution (Fig. 2f). This single *AstC-mTh* neuron, with its soma located in the mesothoracic ganglion, extensively innervates the dorsal regions of the prothoracic and mesothoracic ganglia and sends an ascending projection that arborizes around the SEZ and inferior dorsal brain. The descending processes from this *AstC-mTh*

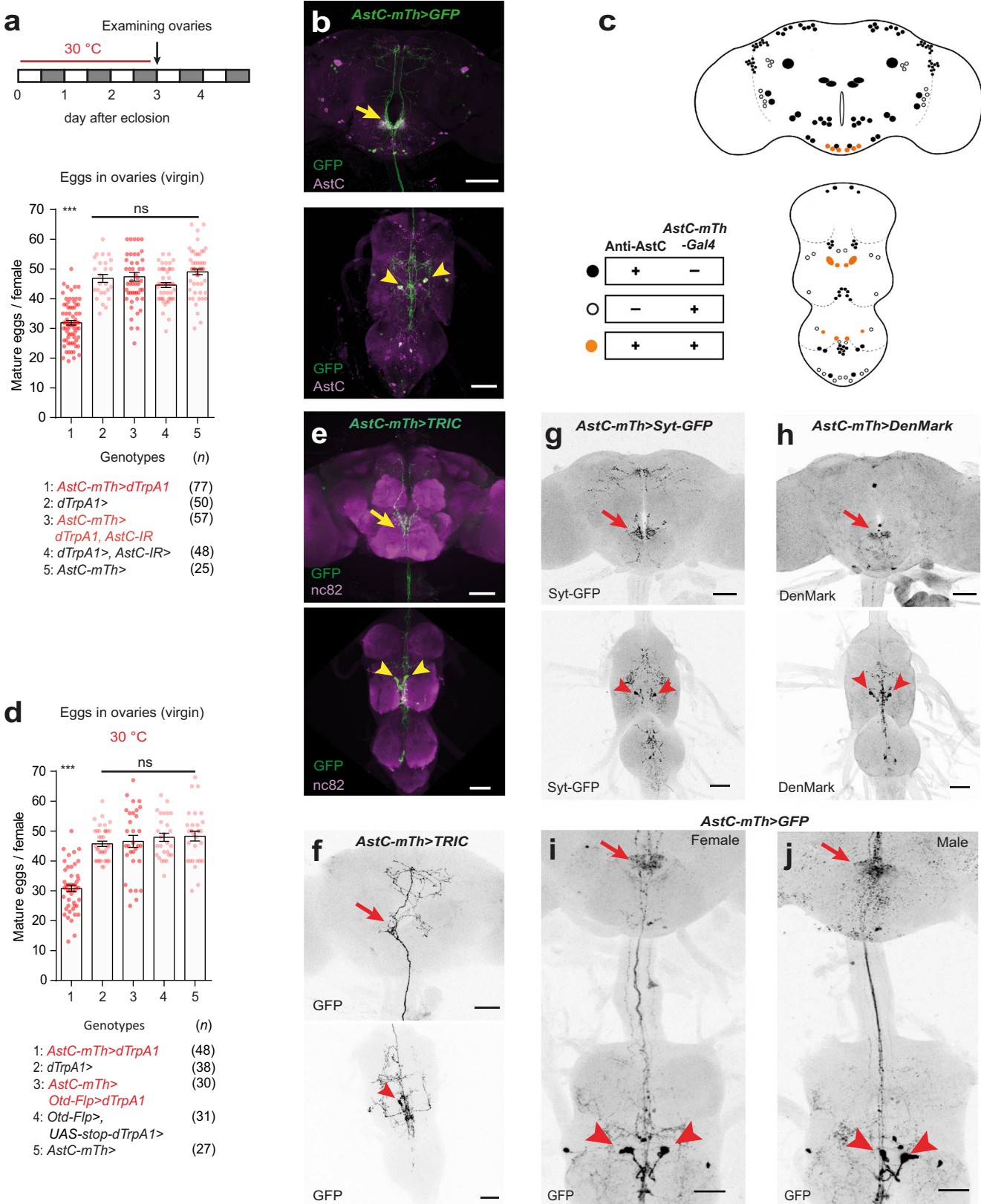

neuron do not project beyond the mesothoracic ganglion to the abdominal ganglion.

Next, by driving the expression of the post-synaptic marker DenMark[30] and the pre-synaptic marker Syt-GFP[31] with *AstC-mTh-Gal4*, we sought to identify, respectively, the inputs and outputs of the *AstC-mTh* neurons. We observed staining of Syt-GFP indicating the outputs of *AstC-mTh* in the dorso-lateral mesothoracic ganglion, SEZ, and inferior dorsal regions of the brain (Fig. 2g). DenMark staining was evident in somas, running medially along the mesothoracic ganglion to the SEZ, suggesting

**Fig. 2 Identification and characterization of *AstC-mTh* neurons. a, d** Above, experimental protocol for **a** and **d**. Bottom, the number of mature eggs per virgin female of the indicated genotypes 3 days after eclosion at 30 °C. One-way ANOVA followed by Tukey's test for multiple comparisons; ***$p < 0.001$; ns (non-significance) $p > 0.05$. Error bars indicate s.e.m. For adjusted $p$ values, see Supplementary Table 1. **b** Confocal Z-projection images of the brain (above) and VNC (bottom) of 4-day-old virgin females carrying *AstC-mTh-Gal4* and *UAS-mCD8-EGFP* stained with anti-AstC (magenta) and anti-GFP (green). Arrowheads indicate two *AstC-mTh* neurons that project into the SEZ (arrow). Scale bars, 50 μm. **c** A schematic indicating anti-AstC and *AstC-mTh-Gal4-positive* neurons (orange circles), *AstC-mTh-Gal4*-positive neurons lacking anti-AstC (open circles), and anti-AstC neurons lacking *AstC-mTh-Gal4* activity (closed circles). **e** Confocal Z-projection images of the brain and VNC of 4-day-old virgin females carrying *AstC-mTh-Gal4* and TRIC transgenes stained with nc82 antibody (magenta) and anti-GFP. TRIC labels two *AstC-mTh* neuron somas in the VNC (arrowheads) that project into the SEZ (arrow). Scale bars, 50 μm. **f** Negative images of TRIC labeling of single *AstC-mTh* neuron (arrowhead) in the CNS of 4-day-old virgin females. TRIC driven by *AstC-mTh-Gal4* labeled one of two *AstC-mTh* neurons. Arrow indicates SEZ region. Scale bars, 50 μm. **g** Negative images of the brain and VNC of 4-day-old virgin females carrying *AstC-mTh-Gal4* and *UAS-Syt-EGFP* stained with anti-GFP. Presynaptic domains labeled by Syt-EGFP are evident in the dorsal SEZ region (arrow). Arrowheads indicate *AstC-mTh* neuron somas. Scale bars, 50 μm. **h** Negative images of the brain and VNC of 4-day-old virgin females carrying *AstC-mTh-Gal4* and *UAS-DenMark* stained with anti-RFP. Postsynaptic domains labeled by DenMark are evident in the dorsal SEZ region (arrow) and *AstC-mTh* neuron somas (arrowheads). Scale bars, 50 μm. **i, j** Negative images of the CNS of 4-day-old virgin female (**i**) and male (**j**) carrying *AstC-mTh-Gal4* and *UAS-myr-EGFP* stained with anti-GFP. Arrow indicates SEZ region. Arrowheads indicate *AstC-mTh* neuron somas. Scale bars, 50 μm. For a detailed list of genotypes, see Supplementary Table 2.

they are *AstC-mTh* neuron inputs (Fig. 2h). The *AstC-mTh* neurons and their ascending projection that arborizes in the SEZ were evident in both sexes (Fig. 2i, j).

**AstC-mTh neurons inhibit JH biosynthesis in the CA.** AstC was initially identified due to its direct allatostatic actions on the CA of the hawkmoth *M. sexta*[21]. Therefore, we next asked whether *AstC-mTh* neurons inhibit JH production. We measured JH-III levels from female whole-body extracts (Fig. S3). We found that $w^{1118}$ females produced ~10 times higher JH-III levels on the first-day post-eclosion (day 1) than during the pupal S8 stage (day -2). Next, we subjected females expressing dTrpA1 in *AstC-D-Gal4* neurons to thermal activation for three days beginning at pupal stage S8 and then measured JH-III levels on day 1 post-eclosion. Indeed, we found thermal activation of *AstC-D-Gal4* neurons produced a significant suppression of JH production. To further understand the role of *AstC-mTh* neurons in modulating JH titers, we evaluated JH signaling by quantifying mRNA levels of *Krüppel homolog 1* (*Kr-h1*), a transcriptional target of JH signaling[32]. In our previous study, we confirmed that Kr-h1 mRNA levels mirror JH levels with good sensitivity and fidelity[23]. When we activated *AstC-D-Gal4* neurons, we observed a significant reduction in Kr-h1 transcript, to a level ~75% of that of the control group (Figs. 3a and S3). To provide further evidence, we asked how the JH mimic methoprene affects vitellogenesis in the presence of *AstC-D-Gal4* neuron activation. We found females treated with methoprene showed a significant de-repression of the inhibition of total oogenesis induced by a 3-day thermal activation of *AstC-D-Gal4* neurons (Fig. 3b).

Because activation of *AstC-mTh* neurons seems to inhibit JH biosynthesis, we next tested the possibility that the JH-producing CA receives the AstC signal directly. The *Drosophila* genome contains two AstC-receptor genes, *star1* (aka *AstC-R1*) and *AICR2* (aka *AstC-R2*), both of which encode G-protein coupled receptors highly sensitive to and selective for AstC[33]. We examined the CA in *AstC-R1-Gal4* or *AstC-R2-Gal4* females, each of which carries an extra exon of the TA-Gal4 transgene in their respective receptor loci[23,34]. Juvenile Hormone Acid O-Methyl Transferase (JHAMT) is the rate-limiting enzyme for JH production, and it is expressed exclusively in the CA[35]. We found the CA labeled by anti-JHAMT were also positive for both *AstC-R1-Gal4* and *AstC-R2-Gal4* (Fig. 3c, d). Finally, when we knocked down each receptor in the CA one at a time, we found depletion of each receptor accelerated and increased vitellogenesis in young virgin females, recapitulating the phenotype we observed in those with AstC deficiency or *AstC-mTh* neuronal silencing (Fig. 3e). When we used *JHAMT-Gal4* to drive over-

expression of either AstC-R1 or -R2 in the CA, we saw a limited effect on total oogenesis, as measured by the number of mature stage 14 eggs. But simultaneous expression of both AstC-R1 and -R2 in the CA significantly reduced total oogenesis (Fig. 3f). Together, these results suggest the CA is indeed exposed to AstC during reproductive maturation.

Our genetic and biochemical evidence that associates the *AstC* gene and the *AstC-mTh* neurons with JH production is compelling, but another recent study implicated AstC from gut enteroendocrine cells in promoting feeding in females[36]. Thus, we wondered whether the precocious vitellogenesis of AstC-deficient females is associated with their altered feeding activity. To test this possibility, we examined the feeding behavior of female flies with AstC deficiency or silenced *AstC-mTh* neurons during and after reproductive maturation using an automated feeding activity monitor[37]. For 24 h after eclosion, both AstC-deficient mutant and control females displayed very low feeding with no significant difference in any of the examined feeding parameters (Fig. S4a). Thus, it is unlikely that AstC deficiency induces precocious vitellogenesis indirectly by promoting feeding activity. In fully mature females (i.e., 4–5 days after eclosion), AstC deficiency moderately reduced mean feeding duration, but the silencing of *AstC-mTh* neurons did not (Fig. S4b, c). Thus, *AstC-mTh* neurons are unlikely involved in regulating feeding. Consistently, we examined TRIC signal in the *AstC-mTh* neurons of flies under opposing feeding conditions (i.e., starvation vs. ad libitum feeding for 24 h), failing to detect any measurable difference between them (Fig. S4d).

**AstC-mTh neurons inhibits CA activity likely via a hormonal route.** To determine whether *AstC-mTh* neurons innervate the CA directly, we expressed *UAS-myr-EGFP* in *AstC-mTh-Gal4* neurons. We did not detect any of the resulting GFP signal around the CA of females (Fig. S5a). It is also unlikely that other AstC-positive neurons innervate the CA, because we did not find any anti-AstC labeling in projections around the CA (Fig. S5b). Thus, any communication taking place between *AstC-mTh* neurons and the CA is unlikely to occur via a neuronal route. We did, however, notice *AstC-mTh* projections in the dorsomedial SEZ where the aorta contacts the brain (arrow in Fig. 2f, g). This is reminiscent of the SEZ innervation of insulin-producing cells (IPCs), which are neuroendocrine cells that also stimulate JH biosynthesis[38,39]. In a previous study, we found that the activation of another subset of AstC neurons (i.e., *AstC-DN1p* neurons) inhibits JH biosynthesis by suppressing the secretory activity of the IPCs, causing a significant increase in anti-Dilp2 staining as Dilp2 accumulates[23]. For this reason, we tested the possibility that

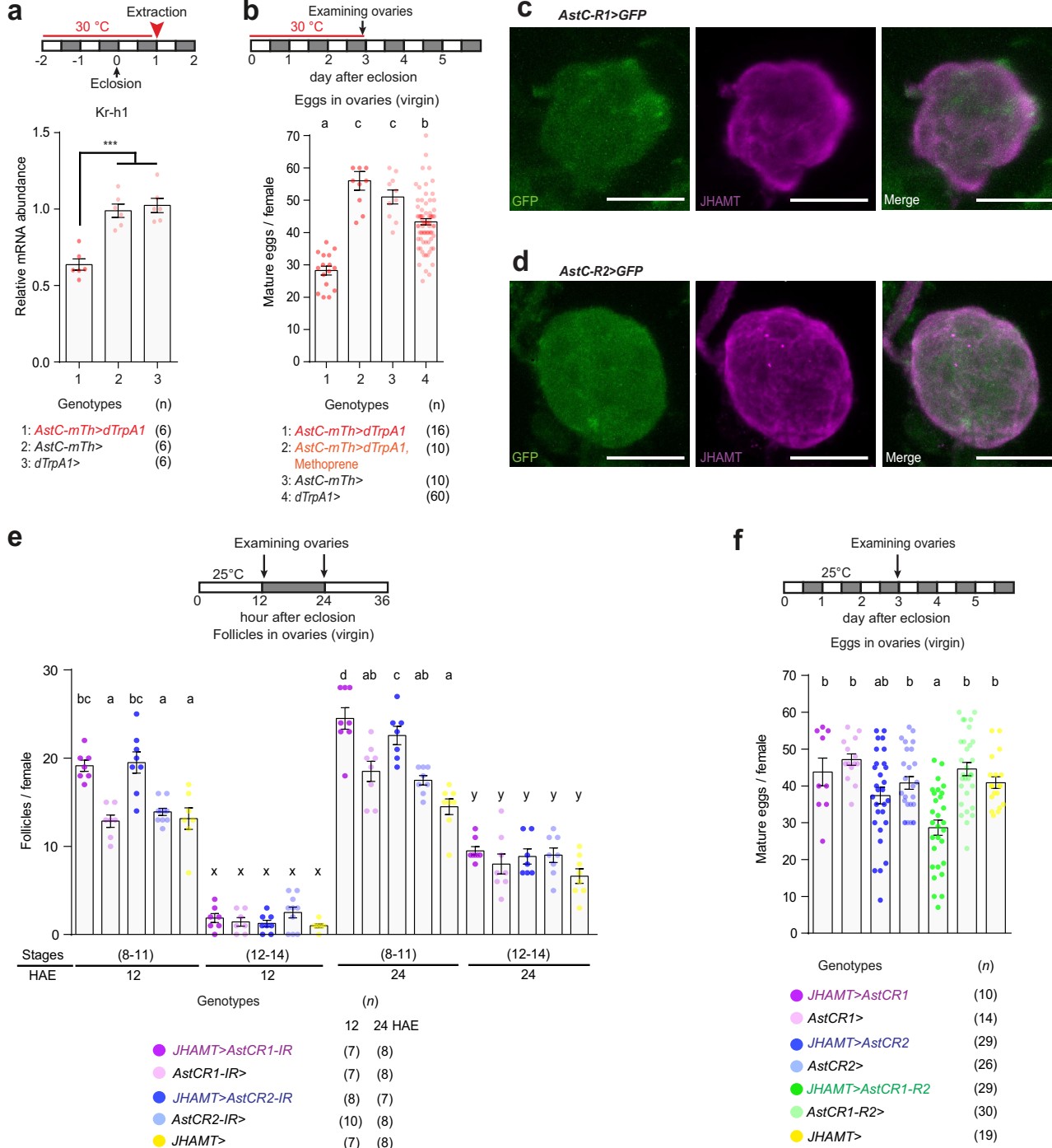

**Fig. 3 *AstC-mTh* neurons regulate JH biosynthesis via AstC-R1 and AstC-R2 in the CA. a** Above, experimental protocol. Below, Kr-h1 transcript levels in adult females of the indicated genotypes maintained for 3 days at 30 °C. One-way ANOVA followed by Tukey's test for multiple comparisons; ***$p < 0.001$. Data is also shown in Fig S3b. Error bars in **a**, **b**, **e**, and **f** indicate s.e.m. For adjusted $p$ values, see Supplementary Table 1. **b** Above, experimental protocol. Below, number of mature eggs per virgin female of the indicated genotypes 3 days post-eclosion at 30 °C. Genotype 2 was fed methoprene while the others were fed the vehicle control. The letters above the columns indicate significant differences ($p < 0.05$) for comparisons of genotypes (One-way ANOVA followed by Tukey's test for multiple comparisons). **c**, **d** The CA of 4-day-old virgin females carrying *AstC-R1-Gal4* (**c**) or *AstC-R2-Gal4* (**d**) and *UAS-mCD8-EGFP* stained with anti-JHAMT (magenta) and anti-GFP (green). Scale bars, 25 μm. **e** Above, experimental protocol. Below, number of stage 8–11 and 12–14 follicles per virgin female of the indicated genotypes at the indicated number of hours after eclosion (HAE). The letters above the columns indicate significant differences ($p < 0.05$) for comparisons of time points and genotypes (two-way ANOVA with Bonferroni post-hoc test for multiple comparisons); a-d for stage 8–11 and x-y for stage 12–14. **f** Above, experimental protocol. Below, number of mature eggs per virgin female of the indicated genotypes 3 days post-eclosion at 25 °C. The letters above the columns indicate significant differences ($p < 0.05$) for comparisons of genotypes (one-way ANOVA followed by Tukey's test for multiple comparisons). For a detailed list of genotypes, see Supplementary Table 2.

*AstC-mTh* neuron activation inhibits IPC activity, affecting JH biosynthesis indirectly. Unlike *AstC-DN1p* neurons, however, thermal activation of *AstC-mTh* neurons had a limited impact on anti-Dilp2 levels in IPCs (Fig. S6a). Inhibition of IPCs or insulin signaling during development typically leads to adults with small body size. And while constitutive activation of *AstC-DN1p* neurons does indeed produce adults with small body size[23], similar activation of *AstC-mTh* neurons does not measurably affect adult body size (Fig. S6b). Thus, we concluded that unlike *AstC-DN1p* neurons, *AstC-mTh* neurons do not function through the IPCs. Considering the absence of AstC processes innervating the CA, the most parsimonious hypothesis is that *AstC-mTh* neurons, like IPCs, also secrete their contents (i.e., AstC) into the circulation to suppress JH biosynthesis in the CA via a hormonal route.

**AstC-mTh neurons play a role in post-mating vitellogenesis**. In *D. melanogaster*, the mating signal SP stimulates oogenesis progression, specifically vitellogenesis[6]. We reasoned that SP likely does this by inhibiting *AstC-mTh* neurons, relieving AstC-mediated inhibition of the CA. To explore this hypothesis, we first evaluated the activity of *AstC-mTh-Gal4* neurons in females before and after mating using TRIC (Fig. 4a). *AstC-mTh* neurons in 4-day-old virgin females show strong TRIC signal, but this is significantly down-regulated within 48 h after mating with control males, but not after mating with SP-less males.

Because SP appears to inhibit the activity of *AstC-mTh* neurons, we wanted to determine whether SP stimulates vitellogenesis via the *AstC-mTh* neurons. We therefore counted vitellogenic follicles (stages 8–14) and oviposited eggs every four hours after mating for two days (Fig. S7). In control *w[1118]* females, we first observed oviposition activity 4 hours after mating, leading to a concomitant reduction in stage 14 eggs. We were unable to observe mating-induced vitellogenesis until 12 hours post-mating, when the number of stage 10 follicles rose compared with virgin controls. Of note, we did not observe any measurable increase in follicles of other stages, whereas stage 10 follicles remained consistently elevated until the end of the experiment. Since it takes ~12 hours for pre-vitellogenic stage 7 follicles to become stage 10 follicles[40], the increase in stage 10 follicles observed 12 hours post-mating likely reflects mating-induced vitellogenesis commencing almost immediately after mating. Consistent with this interpretation, females mated with SP-less males showed no increase in stage 10 follicles, as examined 24 h post-mating (Fig. S7g).

Next, we asked whether *AstC-mTh* neurons control mating-induced vitellogenesis. Females expressing dTrpA1 in *AstC-mTh* neurons (i.e., *AstC-mTh-Gal4* neurons) were incubated at 30 °C for 24 or 48 h after mating (Figs. 4b and S8a). We assumed that thermal activation of *AstC-mTh* neurons would override SP-induced inhibition of *AstC-mTh* neurons, restoring virgin-like *AstC-mTh* activity. Indeed, thermal activation blocked mating-induced vitellogenesis, reducing stage 10 follicles by ~50% when compared with controls (Fig. 4b). We also expected *AstC-mTh* neuronal silencing would recapitulate mating-induced vitellogenesis even in virgin females. When we blocked the secretory activity of *AstC-mTh* neurons via expression of *Shi[ts]*, we did not observe any difference in the number of vitellogenic follicles (stages 8–14) in 4-day-old virgin females (Fig. S8b). Thus, *AstC-mTh* neurons do not seem to stimulate vitellogenesis per se, but instead gate the vitellogenesis progression that is stimulated by other mating-associated allatotropic factors. Ecdysis triggering hormone (ETH), which was originally discovered as an obligatory molting factor, is one allatotropic that promotes JH production and reproduction. In the fruit fly, ETH expression and secretion depend on ecdysone (20E)[41]. In post-mating females, SP induces 20E biosynthesis via the neuronal SP response pathway[7]. Therefore, we asked whether ETH signaling promotes post-mating vitellogenesis by knocking-down ETH receptor (ETHR) in the CA. As with *AstC-mTh* neuron activation, *ETHR-RNAi* caused a ~50% decrease of stage 10 follicles compared with controls (Fig. 4c). Thus, we propose that the SP signal stimulates post-mating vitellogenesis by simultaneously elevating ETH-induced allatotropic activity and relieving AstC-induced allatostatic activity (Fig. 4e).

*AstC-mTh* neurons are interneurons with neurites that innervate the CNS. Because SP inhibits *AstC-mTh* neurons in the same way it inhibits SAG neurons and other neural components of the SP response circuit[15,16], we wanted to determine whether *AstC-mTh* neurons are functionally linked with SAG neurons. Thus, we prepared female flies expressing TrpA1 in SAG neurons and incubated them at 30 °C for 24 h after mating. We expected thermal activation of SAG neurons to override their SP-induced inhibition, thus restoring virgin-like neural activity. Remarkably, we found thermal activation of SAG neurons precisely recapitulated the phenotype induced by thermal activation of *AstC-mTh* neurons we described above; it blocked mating-induced vitellogenesis and reduced the number of stage 10 follicles by ~50% (Fig. 4d). Unlike with *AstC-mTh* neurons, however, activation of SAG neurons also suppressed oviposition, increasing stage 14 eggs in the ovaries. Thus, SAG neurons modulate both vitellogenesis and oviposition, whereas *AstC-mTh* neurons gate vitellogenesis without affecting oviposition.

**The role of SAG neurons in vitellogenesis during reproductive maturation**. Having shown that SAG neurons and *AstC-mTh* neurons are functionally associated in mating-induced vitellogenesis, we wondered whether SAG neurons are also involved in vitellogenesis in young virgin females during reproductive maturation. When we suppressed SAG neuronal activity with Kir2.1 and examined vitellogenesis at 12 and 24 h after eclosion, we found significantly elevated numbers of early vitellogenic follicles (stages 8–11) just like we observed in females with silenced *AstC-mTh* neurons (compare Figs. 5a and 1e). Next, we asked whether activation of SAG neurons during reproductive maturation suppresses vitellogenesis and reduces total oogenesis. We found thermal activation of SAG neurons for 3 days post-eclosion reduced the number of stage 14 eggs by ~50% (Fig. 5b).

We then wondered whether SAG neurons function through AstC neuropeptide or *AstC-mTh* neurons. In the absence of a functional AstC allele, thermal activation of SAG neurons failed to suppress total oogenesis (Fig. 5b). Having shown that AstC is required for the function of SAG neurons, we wondered whether *AstC-mTh* neurons also function downstream of SAG neurons. To test this hypothesis, we activated SAG neurons while also silencing *AstC-mTh* neurons with Kir2.1. As with AstC deficiency, the silencing of *AstC-mTh* neurons also seemed to relieve the vitellogenesis blockade induced by SAG neuron activation, restoring total oogenesis to control levels (Fig. 5c). Next, we performed the converse experiment in which we simultaneously silenced SAG neurons while activating *AstC-mTh* neurons. As expected, activation of *AstC-mTh* neurons countered the vitellogenesis-stimulating effect of SAG neuron silencing and restored early vitellogenic follicles to control levels (Fig. 5d). On the strength of this evidence, we conclude that *AstC-mTh* neurons function downstream of SAG neurons.

We then asked whether a JH mimic could override SAG neuron activation and restore total oogenesis to control levels. We found methoprene treatment reversed the vitellogenesis

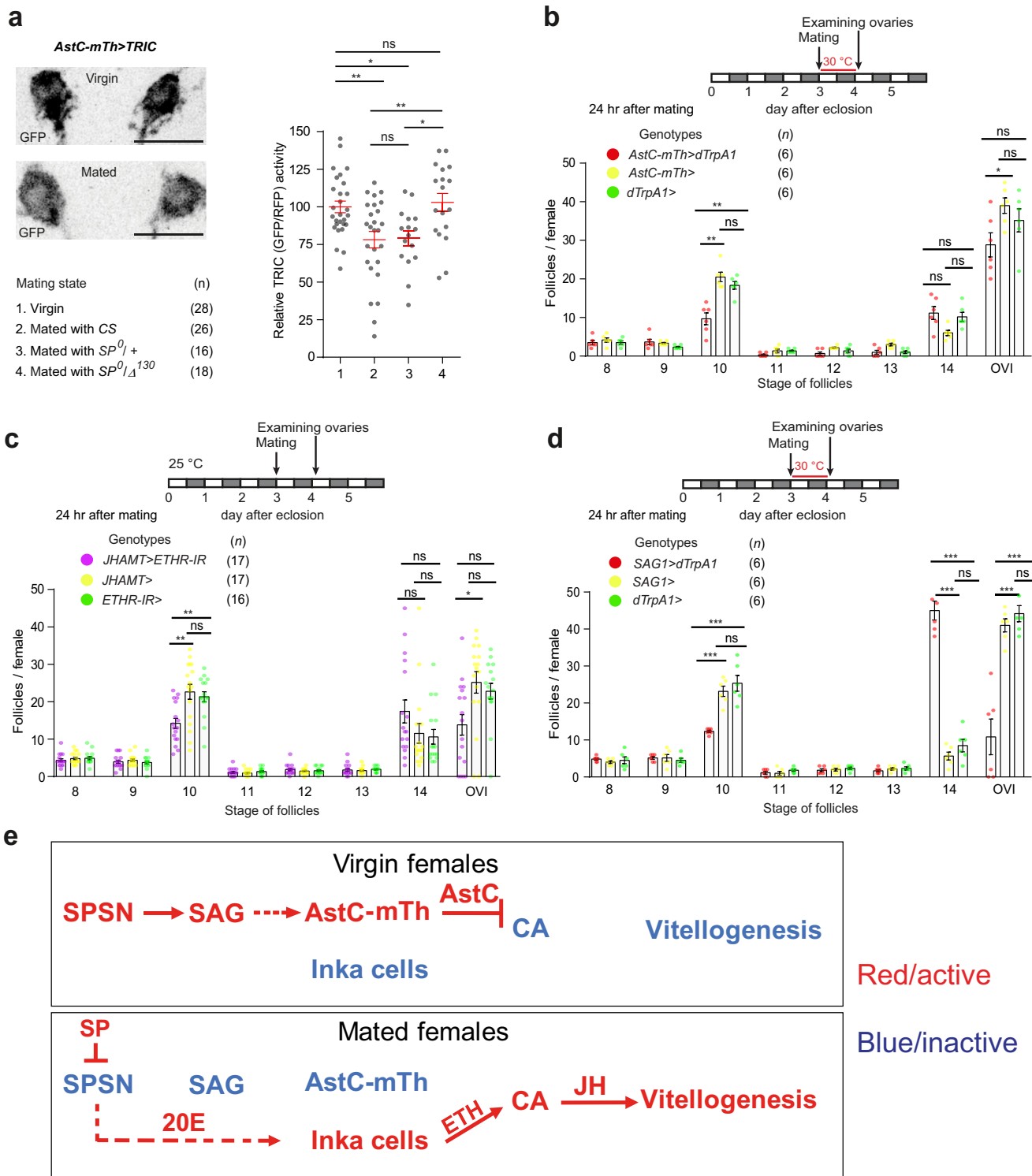

suppression imposed by SAG neuron activation and restored the number of stage 14 eggs to control levels (Fig. 5b). This result provided further evidence of a causal link between SAG neurons and the JH pathway. Finally, we examined SAG neuron activity during reproductive maturation using TRIC analysis (Fig. 5e). As expected, we found evidence of a gradual increase in SAG neural activity the first two days post-eclosion, reaching maximum activity on day 3 when the vitellogenesis associated with reproductive maturation ends (Figs. 1a and 5f).

**SAG neurons seem to connect indirectly with *AstC-mTh* neurons.** Having shown that *AstC-mTh* neurons function downstream of SAG neurons, we wondered how SAG neurons and *AstC-mTh* neurons interact with one another at the cellular level. By labeling the pre-synaptic terminals of SAG neurons with Syt-GFP, we determined that SAG outputs arborize near *AstC-mTh* somas and in the mediodorsal SEZ where *AstC-mTh* neurons project and form extensive post-synaptic terminals (Fig. S9a). Thus, we further explored the possibility that SAG neurons form functional synapses with *AstC-mTh* neurons by adopting the

**Fig. 4 AstC-mTh neurons gate mating-induced vitellogenesis in response to SP. a** Left, negative images of TRIC labeling in *AstC-mTh* neurons of virgin (left) and mated (right) females carrying *AstC-mTh-Gal4* and TRIC transgenes, indicating intracellular $Ca^{2+}$ transients. Scale bars, 10 µm. Right, relative TRIC (GFP/RFP) intensities from TRIC-expressing *AstC-mTh* neurons of virgin females and females mated with males of the indicated genotypes. One-way ANOVA followed by Tukey's test for multiple comparisons among genotypes; $**p < 0.01$; $*p < 0.05$; ns (non-significance) $p > 0.05$. Error bars indicate s.e.m. For adjusted $p$ values, see Supplementary Table 1. **b–d** Above, experimental protocol. Below, number of follicles of the indicated stage and oviposited eggs (OVI) from females of the indicated genotypes 24 h after mating. For the thermal activation experiments (**b, d**), females were incubated at 30 °C for 24 h after mating. One-way ANOVA followed by Tukey's test for multiple comparisons among genotypes; $***p < 0.001$; $**p < 0.01$; $*p < 0.05$; ns (non-significance) and no labeling, $p > 0.05$. Error bars indicate s.e.m. For a detailed list of genotypes, see Supplementary Table 2. **e** A model that explains how SP induces post-mating vitellogenesis via a neuronal pathway composed of SPSN and SAG neurons. SPSN and SAG neurons are electrically active in mature virgin females. Active SAG neurons indirectly stimulate *AstC-mTh* neurons (arrow with dotted line) to secrete AstC into circulation. Circulating AstC then inhibits JH biosynthesis in the CA and sets up a vitellogenesis blockade. Upon mating, SP silences SPSN and SAG neurons in sequence. This inhibition relieves the AstC-induced vitellogenesis blockade. At the same time, SP also elevates 20E via SPSN. This 20E stimulates the endocrine Inka cells to secrete ETH, which induces JH biosynthesis in the CA. Thus, the neuronal SP pathway stimulates post-mating vitellogenesis in mated females by elevating circulating allatotropin (i.e., ETH) and reducing circulating allatostatin (i.e., AstC).

*trans*-Tango technique, which visualizes functional post-synaptic neurons[42]. We found SAG neuron expression of *trans*-Tango labeled many post-synaptic neurons, primarily in the abdominal ganglion. But even after many attempts, we failed to detect *trans*-Tango activity in *AstC-mTh* neurons (Fig. S9b). Thus, it seems unlikely that SAG neurons form conventional chemical synapses with *AstC-mTH* neurons.

We further probed the possibility that SAG neurons provide functional inputs to *AstC-mTh* neurons through non-synaptic transmission. Specifically, we measured the membrane potential of *AstC-mTh* neurons via whole-cell patch clamping while activating SAG neurons via optogenetics (Fig. S10). We activated CsChrimson-expressing SAG neurons with ~610-nm light for one second via a train of step pulses with a period of 50 ms and a 50% duty cycle (Fig. S10a–c). We did not, however, observe any noticeable change in *AstC-mTh* neuron membrane potential upon activation of SAG neurons in 2–3-day-old mated female flies (Fig. S10d, f, g). Notably, the same activation protocol induced strong light-evoked neuronal depolarization in a sample SAG cell (Fig. S10e). This suggests the influence of SAG neurons on *AstC-mTh* neurons we observed in our molecular genetic experiments occurs via long-term, slow, and indirect modulation. Thus, we propose that there are additional unknown neuron(s) that connect SAG and *AstC-mTh* neurons, which seem to require long-term excitatory inputs to relay a signal to the *AstC-mTh* neurons.

## Discussion

Vitellogenesis initiation is a critical control point for oogenesis in *D. melanogaster*. In this species, vitellogenesis begins shortly after eclosion and continues through reproductive maturation, during which females prepare for mating and egg-laying. As females mature over two or three days, vitellogenesis ceases until mating stimulates it again to sustain egg-laying activity. The seminal substance SP acts as a mating signal, stimulating vitellogenesis, ovulation, and oviposition. In this study, we identified a pair of thoracic ganglion neurons (i.e., *AstC-mTh* neurons) that express the insect somatostatin (SST) AstC. We provide evidence that AstC from these neurons modulates vitellogenesis by gating two distinct episodes of JH biosynthesis, one stimulated by adult molting and the other stimulated by mating. This finding highlights a striking conservation in the AstC/SST signaling system between insects and mammals. Not only are the mammalian SST receptors (sstr1-5) orthologous to the *Drosophila* AstC receptors, but just as AstC inhibits the insect gonadotropin JH, SST inhibits the mammalian gonadotropins follicle-stimulating hormone (FSH) and leutenizing hormone (LH) via the hypothalamic neuropeptide gonadotropin-releasing hormone (GnRH)[43].

**AstC-mTh neurons delay reproductive maturation**. In *D. melanogaster*, JH acts on the ovary to initiate vitellogenesis[44,45]. JH levels peak at eclosion (day 0) and decrease gradually over several days[46,47]. JH titer and therefore oogenesis progression are closely coupled to the endocrine events that induce eclosion[41,48]. The ETH that induces molting behaviors (i.e., eclosion) also functions as a potent allatotropin[49,50]. When a pharate adult female is ready to molt, the endocrine Inka cells secrete ETH. This enters the circulation and triggers a sequence of stereotypic motor patterns culminating in eclosion. When circulating ETH reaches the CA, it triggers the post-eclosion surge of JH. Considering the 28-h delay required for previtellogenic stage 7 follicles to develop into stage 11 follicles, the ETH-induced JH surge at day 0 is likely responsible for the marked increase of early vitellogenic follicles (stages 8–11) detected at day 1. In this study, we found that AstC deficiency in *AstC-mTh* neurons and silencing *AstC-mTh* neurons both advance vitellogenesis initiation by ~12 h. This suggests AstC delays the JH peak by ~12 h and that *AstC-mTh* neurons are involved in a temporal decoupling of eclosion and reproductive maturation. It is unclear why *Drosophila* has evolved a mechanism to delay reproductive maturation. In addition to vitellogenesis, JH also stimulates other processes associated with reproductive maturation, such as pheromone production and the development of mating receptivity[51]. In the wild, *D. melanogaster* males often wait for females to emerge from their pupal cases before forcefully mating with them[52,53]. We speculate a programmed delay in the processes required for developing attractiveness toward males and mating receptivity would contribute to female fitness by increasing the temporal window in which females can select the best available suitor. The time required for females to reach reproductive maturity varies across *Drosophila* species. For example, *D. pachea* females require weeks to become ready to mate, whereas *D. mettleir* females are ready to mate within hours of eclosion. Thus, a comparative analysis of AstC's role in programming the delays required for reproductive maturation across species would be of great interest[54].

As with AstC deficiency, we found SAG neuron silencing also advanced post-eclosion vitellogenesis by ~12 h. In contrast, activation of SAG neurons during reproductive maturation (i.e., for 3 days post-eclosion) reduced oogenesis by ~50%, precisely phenocopying *AstC-mTh* neuron activation. We also found *AstC-mTh* neuron silencing blocks the oogenesis-suppressing effect of SAG neuron activation. This epistatic relationship between SAG neurons and *AstC-mTh* neurons strongly supports the hypothesis that *AstC-mTh* neurons function downstream of SAG neurons. Using the TRIC technique, we found a gradual increase in intracellular $Ca^{2+}$ in SAG neurons over the 3 days following eclosion, indicating increased activity. This is consistent with the significant levels of spontaneous firing observed in patch

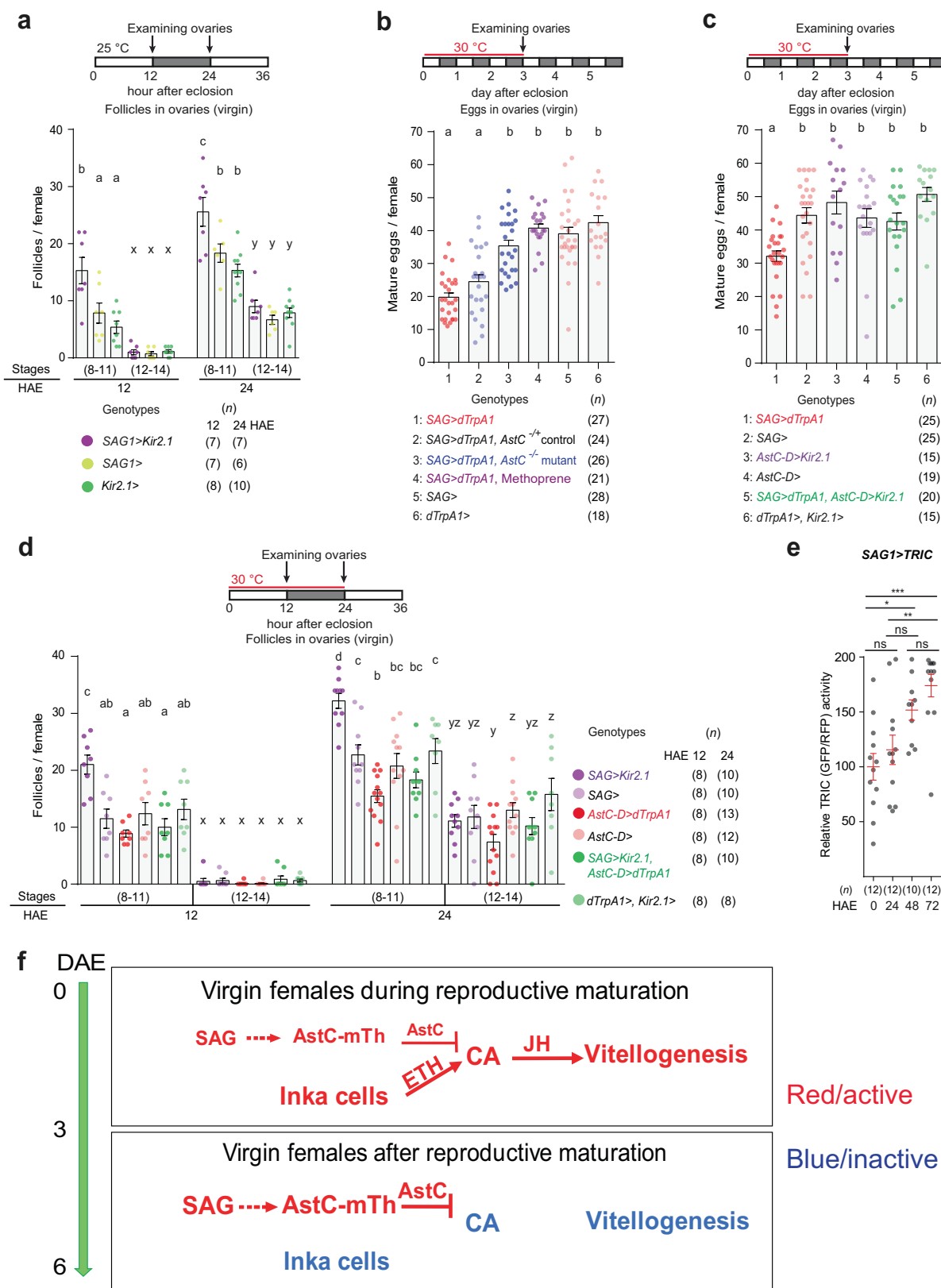

recordings from the SAG neurons of 4–5-day-old virgin females[16]. We propose that as females undergo reproductive maturation, SAG neurons augment the excitatory inputs into *AstC-mTh* neurons, driving them to secrete more AstC and cause further inhibition of JH biosynthesis (Fig. 5f).

Although we found AstC deficiency significantly advanced vitellogenesis, it had a limited effect on total oogenesis in virgin females. This is probably because virgin females have limited pre-vitellogenic follicles that can enter vitellogenesis. Consistent with this interpretation, JH has no effect on GSC proliferation[7]. In

**Fig. 5 SAG neurons function upstream of *AstC-mTh* neurons and gate vitellogenesis during reproductive maturation. a, d** Above, experimental protocol. Below, number of stage 8–11 and 12–14 follicles per virgin female of the indicated genotypes at the indicated hours after eclosion (HAE). The letters above the columns indicate significant differences ($p < 0.05$) for comparisons of time points and genotypes (two-way ANOVA with Bonferroni post-hoc test for multiple comparisons); **a–d** for stage 8–11 and x–z for stage 12–14. For adjusted $p$ values, see Supplementary Table 1. Error bars in **a–e** indicate s.e.m. To target SAG neurons, we used *SAG1-Gal4*, which combines the *VT50405-p65AD* and *VT7068-GAL4DBD* transgenes. We also used *SAG-LexA* (i.e., *VT50405-LexA*) or *SAG-Gal4* (i.e., *VT50405-Gal4*). Although the CNS expression driven by *SAG1-Gal4* is slightly more restricted than that of the *VT50405* transgene alone (i.e., *SAG-LexA* or *SAG-Gal4*)[16], *SAG1-Gal4* or *SAG-LexA* induced Kir2.1 expression produced similar vitellogenesis phenotypes (compare the magenta circles in **a** and **d**). **b, c** Above, experimental protocol. Below, number of mature eggs per virgin female of the indicated genotypes 3 days after eclosion under 30 °C. The letters above the columns indicate significant differences ($p < 0.05$) for comparisons of genotypes (one-way ANOVA followed by Tukey's test for multiple comparisons). **e** Relative TRIC (GFP) intensities from SAG neurons of TRIC virgin females showing $Ca^{2+}$ activity at the indicated HAE. One-way ANOVA followed by Tukey's test for multiple comparisons among genotypes; ***$p < 0.001$; **$p < 0.01$; *$p < 0.05$; ns (non-significance) and no labeling, $p > 0.05$. For a detailed list of genotypes, see Supplementary Table 2. **f** A model that explains how endocrine and neuronal events stimulate and terminate vitellogenesis during reproductive maturation. During adult molting, the Inka cells secrete ETH to trigger eclosion behaviors. The resulting increase in circulating ETH induces JH biosynthesis and generates a post-eclosion JH surge. This stimulates vitellogenesis shortly after eclosion. As female completes reproductive maturation, circulating ETH levels fall and SAG neurons augment the activation of *AstC-mTh* neurons, driving them to secrete AstC. This then inhibits JH biosynthesis and terminates vitellogenesis.

---

mated females, SP stimulates the ovary to produce ecdysteroids, which in turn, stimulate GSC proliferation and presumably pre-vitellogenic follicles.

***AstC-mTh* neurons in mating-induced vitellogenesis**. An ex vivo analysis found that synthetic SP can induce JHB3 biosynthesis in isolated CA from 3–4-day-old virgin females[55]. Subsequently, SP was implicated in mating-induced vitellogenesis characterized by a pronounced increase of vitellogenic stage 10 follicles[6]. When SP from the male ejaculate was detected in the hemolymph of mated females, it was proposed to enter the circulation and hormonally stimulate the CA[56]. More recent studies since the discovery of SPR, however, have suggested SP acts primarily through neuronal pathways comprising SPSNs and SAG neurons[16,18]. Of particular note, targeted expression of membrane-tethered SP (mSP) in uterine SPR neurons (i.e., SPSNs) induces virgin females to lay a large number of unfertilized eggs[12,13]. In other words, when mSP can activate SPR only in the neurons that express it rather than entering circulation, virgin female egg-laying resembled that of mated females. Thus, it is likely that mSP stimulates vitellogenesis exclusively through a neuronal route through SPSNs. Moreover, SP induces 20E biosynthesis via the neuronal SP response pathway[7]. This, in turn, activates ETH expression and secretion from adult Inka cells[41]. In this study, our results indicate post-mating vitellogenesis requires not only the absence of AstC-induced allatostatic activity, but also the presence of ETH-induced allatotropic activity. Thus, we propose that the neuronal SP pathway stimulates JH biosynthesis and vitellogenesis by simultaneously inhibiting *AstC-mTh* neurons and activating ETH secretion (Fig. 4e).

Unlike the SAG neurons, which suppress egg-laying activity, *AstC-mTh* neurons have limited impact on egg-laying. Egg-laying is the outcome of a coordinated array of reproductive processes, such as oogenesis (including vitellogenesis), ovulation, and oviposition. All these processes are triggered by the neuronal SP pathway. For example, the neural circuit that links SAG neurons and oviposition behavior was recently discovered[18]. Activation of these neurons stimulates oviposition only in mated females, not virgin females. This is likely because this circuit is specialized for oviposition and can only function after oogenesis and ovulation. Our work establishes a distinct branch of the neuronal SP pathway that is specialized exclusively for vitellogenesis.

**A source of hormonal AstC that acts directly on the CA**. Since Kramer et al.[21] discovered AstC in the hawkmoth *M. sexta* and

reported that it inhibits JH biosynthesis in isolated CAs, it has remained unclear how AstC regulates JH biosynthesis and vitellogenesis through the CA. In this study, we propose *AstC-mTh* neurons produce AstC, which then circulates in the hemolymph to regulate JH biosynthesis associated with reproductive maturation and the post-mating response. We have presented two major lines of evidence. First, *AstC-mTh* neurons, with their somas in the mesothoracic ganglion, project into the dorsal region of the brain's SEZ. The SEZ is also innervated extensively by other neurosecretory neurons such as the IPCs. This anatomical feature suggests *AstC-mTh* neurons are neurosecretory, secreting their contents into the hemolymph. This would allow AstC to travel outside the CNS to the CA, which expresses two highly sensitive and selective GPCR-type AstC receptors (i.e., star1 and AstC-R2). In a second, more important line of evidence, we found knockdown of either of these receptors precisely recapitulated the 12 hour-advance of vitellogenesis initiation that occurs in females with *AstC-mTh* neurons that lack AstC or that are silenced by Kir2.1.

**A potential link between *AstC-mTh* neurons and post-mating immune responses**. In *D. melanogaster*, mating stimulates the innate immune system and induces the production of diverse antimicrobial peptides (AMP) including Metchnikowin, Diptericin, and Drosomycin, etc.[57]. As with other aspects of the post-mating response, SP is responsible for post-mating AMP induction. Females that mate with males lacking SP do not produce AMPs upon mating, and females that express SP ectopically and constitutively in the fat body produce AMP regardless of their mating status. Genetic evidence suggests SP induces AMP production via the Toll and Imd pathways. Interestingly, AstC was recently found to have an immunosuppressive function, dampening the Imd pathway[58]. Thus, our finding that SP reduces AstC secretion from *AstC-mTh* neurons offers an additional and indirect route by which SP boosts the innate immune response and AMP production.

In this study, we have shown that the insect SST/AstC pathway modulates JH biosynthesis during reproductive maturation and the post-mating response. JH levels are also linked to other processes associated with female reproduction, such as somatic organ remodeling[59], pheromone production, and mating receptivity[51]. Moreover, *AstC-mTh* neurons are also present in males, in which JH regulates processes associated with male reproduction like sex pheromone detection[60] and male accessory gland development[61,62]. Future studies will be necessary to evaluate the roles the *AstC-mTh* neurons play in these diverse reproductive processes.

## Methods

**Fly stocks.** Flies were raised at 25 °C and 60% humidity under a 12 h:12 h light:-dark cycle on standard fly media. The stocks used in this study were previously reported or obtained from the Bloomington Drosophila Stock Center (BDSC), the Vienna Drosophila RNAi Center (VDRC), or the Korean Drosophila Resource Center (KDRC). These include $otd^{FLP}$ [28], $UAS > stop > mCD8GFP$[63], $UAS > stop > dTrpA1$[64], $UAS$-$dTrpA1$[65], $UAS$-$Shi^{ts}$[25], $SAG$ ($VT50405$)-$Gal4$[16], $SAG1$-$Gal4$ ($VT50405$-$p65AD; VT7068$-$GAL4DBD$)[16], $SAG$ ($VT50405$)-$LexA$[16], $nSynb$-$Gal4$ carrying $UAS$-$Dicer2$ and $UAS$-$mCD8$-$GFP$ (gifts from Barry J.Dickson, Janelia Research Campus), $y1 w* P\{UAS$-$myrGFP.QUAS$-$mtdTomato$-$3xHA\}$ $su(Hw)$ $attP8; P\{trans$-$Tango\}attP40$ ($trans$-$Tango$)[42], $AstC$-$R1$-$Gal4$[34], $AstC$-$R2$-$Gal4$[34], $JHAMT$-$Gal4$[35], SP-null males $SP^0/\Delta130$[66], $AstC$-$Gal4$[23] (KDRC stock number 10012), $CNMa$-$Gal4$[23] (KDRC #2605), $AstC^1$ [23] (KDRC #10549), $UAS$-$AstC$[23] (KDRC #10550), $UAS$-$Kir2.1$ ($III$) (a gift from Jan Lab, University of California San Francisco), $UAS$-$Dicer2$ (VDRC stock number, 24648), $UAS$-$mCD8::RFP$, $LexAop2$-$mCD8::GFP;nSyb$-$MI::nlsLexLexADbD;UAS$-$p65AD::CaM$ (BDSC stock number, 61679), $UAS$-$Syt$-$EGFP$ (BDSC #6926), $UAS$-$DenMark$ (BDSC #33061), $UAS$-$Kir2.1$ ($II$) (BDSC #6596), $UAS$-$NaChBac$ (BDSC #9466), $UAS$-$AstC$-$R2$-$IR$ (BDSC #25940), $P\{13XLexAop2$-$IVS0csChrimson.mVenus\}attp2$ (BDSC #55139), $P\{y[+t7.7]$ $w[+mC] = 20XUAS$-$IVS$-$GCaMP6m\}attP40$ (BDSC #42748), $UAS$-$AstC$-$R1$-$IR$ (VDRC stock number, 13560), and $UAS$-$AstC$-$IR$ (VDRC #13772), $UAS$-$ETHR$-$IR$(VDRC #101996).

**Molecular biology.** The $AstC$ $Gal4$ lines were generated by dividing the 5′-upstream region of the AstC coding sequence into five ~1-kb tiling fragments. $AstC$-$A$-$Gal4$ ($II$), $AstC$-$B$-$Gal4$ ($II$), $AstC$-$C$-$Gal4$ ($II$), $AstC$-$D$-$Gal4$ ($II$), $AstC$-$D$-$Gal4AD$ ($III$), $AstC$-$E$-$Gal4$ ($II$), and $AstC$-$Gal4DBD$ ($II$) were prepared in gateway vectors as previously described[67]. Each region was amplified via genomic DNA PCR, cloned into the pENTR vector (Invitrogen), and then recombined into pBPGAL4.2::VP16Uw for Gal4, pBPp65AD::ZPUw for Gal4AD or pBPZpGAL4DBD::Uw for Gal4DBD. Each of the final plasmid DNAs was injected into $w^{1118}$ flies with specific landing sites on the second (VIE-72A) or third chromosome (VIE-49B) using the ΦC31 system. VIE-72A and VIE-29Ba were gifts from Barry J. Dickson, Janelia Research Campus. The genomic fragments and primer sequences used to generate these lines were as follows: $AstC$-$A$-$Gal4$ (CACCtttccacgaatgctatgcaa, ggcgtcggtaaatgagaaaa), $AstC$-$B$-$Gal4$ (CACCatttacc-gacgccaatttca, gaaaagccaacagggtggta), $AstC$-$C$-$Gal4$ (CACCtaccaccctgttggcttttc, aaacacggtcgcttaattcc), $AstC$-$D$-$Gal4$ and $AstC$-$D$-$Gal4AD$ (CACCaccgtgtttgccagga-taat, tctgcatgcaacaggtaagc), $AstC$-$E$-$Gal4$ (CACCtgttgcatgcagatgatt, tact-caccggtcctgtttcg), and $AstC$-$Gal4DBD$ (CACCtttccacgaatgctatgcaa, tactcaccggtcctgtttcg). $AstC$-$mTh$-$Gal4$ consists of two split-gal4 transgenes, $AstC$-$Gal4$-$DBD$ and $AstC$-$D$-$Gal4$-$AD$. $UAS$-$AstC$-$R1$ and $UAS$-$AstC$-$R2$ were generated by cloning the NotI-AstC-ORF-KpnI fragment amplified from cDNA clone RH36507 (AY070699) into the SST13 vector and then inserting it into a specific site on the third chromosome (VIE-49B) using the ΦC31 system. The genomic fragments and primer sequences used to generate these lines were as follows: $UAS$-$AstC$-$R1$ ($ttagcggcgcaccatgtttacgtggctgatgat$, $taatctagattacaatctgtctgctgca$) and $UAS$-$AstC$-$R2$ ($aatgcggccgccaccatggaaggtggatggtgg$, $taatctagattataagtccgtgtggagcac$).

**Bioassays.** All assays were repeated on at least two different days. To evaluate vitellogenesis progression after eclosion and total oogenesis, freshly eclosed females were placed individually in vials for the indicated times or for three days and their ovaries were dissected in phosphate-buffered saline (PBS). Vitellogenic follicles (stages 8–14) or mature eggs (stage 14) in both ovaries were counted under a stereomicroscope. The ovarioles were separated and the stages of the vitellogenic follicles were determined according to the method used by Jia et al.[40]. To evaluate mating-induced vitellogenesis progression, freshly eclosed females were aged in groups of 10, mated individually with $Canton$-$S$ males or $SP^0/\Delta130$ males in a 1-cm diameter chamber, and placed individually into vials for the indicated times or for 24 h after mating. Vitellogenic follicles (stages 8–14) and oviposited eggs were counted. For the thermal activation or silencing experiments, virgin or mated females kept individually in vials were transferred into a 30 °C incubator for 24 h immediately after copulation.

**Methoprene treatment.** Methoprene (Sigma-Aldrich, catalog number 40596-69-8) was dissolved in 95% ethanol (1 μg/μl) and added to warm fly food medium (in a liquid state) to produce a final concentration of 1.04 μl/ml[23,68]. Flies were fed this food containing 1.04 μl/ml methoprene or 1 μl/ml 95% ethanol (vehicle control) individually for 3 days after eclosion.

**Immunohistochemistry.** Each fly CNS was dissected in PBS and fixed in 4% paraformaldehyde for 20–30 min at room temperature. After washing with PBST, the samples were then incubated with rabbit anti-GFP antibody (1:1000; Invitrogen, A11122), mouse anti-GFP (1:1000; Invitrogen, A11120), Rabbit anti-DsRed (1:1000; Clontech, 632496), rabbit anti-AstC antibody[23] (1:1000; A gift from Dušan Žitňan, Slovak Academy of Sciences), rat anti-HA antibody (1:100; Roche, 11867423001)[69], rabbit anti-JHAMT[70] (1:1000; a gift from Ryusuke Niwa, University of Tsukuba), anti-Dilp2[71] (1:1000; a gift from Yu Kweon from Korea Research Institute of Bioscience and Biotechnology), and mouse anti-nc82

antibody (1:50; Developmental Studies Hybridoma Bank) for 48 h at 4 °C. Alexa Fluor 488-labeled goat anti-rabbit IgG (1:1000; Invitrogen, A11008), Alexa Fluor 488-labeled goat anti-mouse IgG (1:1000 dilution; Invitrogen, A11001), Alexa Fluor 568 goat anti-rabbit IgG (1:1000 dilution; Invitrogen, A11011), Alexa Fluor 568 goat anti-mouse IgG (1:1000; Invitrogen, A11004), and Alexa Fluor 633 goat anti-rat IgG (1:500; Invitrogen, A21094) were used as secondary antibodies for 24 h at 4 °C. Confocal images were acquired with a Zeiss LSM700/Axiovert 200 M (Zeiss) with Zeiss Zen 2009 software and processed in Image J[72].

To quantify anti-Dilp2 fluorescence, we used a method that was previously described[23]. The maximum intensity Z-projections of 15 consecutive confocal stacks (each 5-μm-thick) covering the somas of all 14 IPCs were merged in Image J. The relative anti-Dilp2 fluorescence intensity of each brain was calculated by setting the average of control brains ($UAS$-$dTrpA1$>) (for Fig S6a) to 100%.

**TRIC analysis.** $AstC$-$mTh$-$Gal4$ and $SAG1$-$Gal4$ were crossed with the TRIC transgenes ($UAS$-$mCD8::RFP$, $LexAop2$-$mCD8::GFP;nSyb$-$MI::nlsLexADBDo;UAS$-$p65AD::CaM$; see above) to quantify changes in intracellular $Ca^{2+}$ in $AstC$-$mTh$ neurons or SAG neurons. CNS tissues were processed as described above (see the Immunohistochemistry section), but not stained with anti-GFP. The only exception is the experiment described in Fig. 4a, in which the tissues were stained with anti-GFP. To quantify EGFP or RFP fluorescence, maximum intensity Z-projections of 15 consecutive 3.82-μm-thick sections covering the entire soma of each neuron were merged in Image J. Then, the value of GFP/RFP of each soma was measured in Image J.

**RT-qPCR.** Total RNA was extracted from whole adult female bodies ($n = 15$) using Trizol (Takara) according to the manufacturer's instructions. RNA (1 μg) was reverse transcribed with oligo (dT) primers (Promega) and Accupower RT premix (Bioneer). Quantitative RT-PCR reactions were performed in 10 μl reaction volumes using a StepOnePlus Real-Time PCR system (Applied Biosystems) with the SYBR Premix Ex taq (Takara) according to the manufacturer's instructions. The gene specific primers for $Rp49$ and $Kr$-$h1$ were described previously[23]: $Rp49$ (forward/reverse, 5′-gacgcttcaagggacagtatctg/5′-aaacgcggttctgcatgag) and $Kr$-$h1$ (5′-gcccaaatatgaatccgctctacc/5′-gtcgtcgcccttgttcatgta).

**JH measurement.** For JH-III quantification, 10 pupa or adult female flies from the indicated genotypes and stages were homogenized with a glass musher and transfer to a silanized vial. Flies homogenates were processed by the method described by Bergot et al.[73] that includes an acetonitrile/pentane extraction and a $C_{18}$ solid-phase extraction cartridge purification. The recovered organic fraction was reduced to a volume of a 100 μl and the JH III epoxide ring was opened by the addition of 150 μl of sodium sulfide and incubation at 55 °C for 30 min. Samples were then extracted with hexane; the recovered organic phase (~500 μl) was filtered with a Nalgene filter (0.2 μm nylon membrane, #176), dried under N2 and stored at −20 °C until used. JH titers from these whole-body extracts were determined using a high performance liquid chromatography coupled to a fluorescent detector protocol (HPLC-FD)[74].

**Wing size measurement.** To measure fly wing size, we modified a method that was previously described[23]. Briefly, the right wings of 3-day-old female flies were dissected and mounted in 70% glycerol. Images of each wing were taken using a LEICA EZ4E and the LAS V4.10 software. The distance between wing landmarks #5 and #13 was measured with ImageJ and used as an indicator of wing size.

**Viability assay.** Three freshly eclosed females and 5 freshly eclosed $CS$ males were incubated at 25 °C in vials with standard media. On day 2, the flies were transferred to new vials to collect eggs. On day 3, the flies were removed and the number of eggs in each vial was counted. The number of pupae and adults that then emerged in each vial was counted on days 7 and 11, respectively. The pupation rate was calculated by dividing the number of pupae by the number of eggs. The eclosion rate was calculated by dividing the number of adults by the number of pupae.

**Feeding assay.** To measure feeding, we used the Fly Liquid-Food Interaction Counter (FLIC) assay[37]. The FLIC master control unit and $Drosophila$ Feeding Monitors (DFM, Sable Systems) were kept at 25 °C and 70% humidity under a 12 h:12 h light:dark cycle. Freshly eclosed females were aged in groups of 10 and loaded individually into the DFM unit with a 5% sucrose solution. Feeding behaviors were recorded for 30 minutes or 24 hours. All recordings started at ZT8. Data were analyzed with custom code written in MATLAB.

**Electrophysiology and optogenetics.** Flies 0-to-12-h post eclosion were transferred to a vial containing the cornmeal food with 0.5-mM all-trans-retinal. The flies were kept in the dark for two days, and only females were prepared for whole-cell patch clamp experiments. For dissection, all legs were removed with the cuticle on the ventral part of the thorax. While leg and haltere nerves were severed, the cervical and abdominal nerves were left intact. A dissected fly was pinned down with the ventral side up on a Sylgard plate and placed under the microscope (Fig. S10a). The glial sheath covering the target cell area was cleared by an

extracellular solution containing 0.5 mg/mL Collagenase (Worthington, USA). We perfused an extracellular solution (275–280 mOSM) that contained in mM: 103 NaCl, 3 KCl, 5 N-Tris(hydroxymethyl) methyl-2-aminoethanesulfonic acid (TES), 10 Trehalose, 10 Glucose, 2 Sucrose, 26 NaHCO3, 1 NaH2PO4, 1.5 CaCl2, 4 MgCl2. Patch-clamp electrodes (4–8 MΩ) were pulled (P-1000, Sutter Instrument), fire-polished (MF-900, Narishige), and then filled with solution containing (in mM): 140 potassium aspartate, 1 KCl, 10 HEPES, 1 EGTA, 0.5 Na3GTP, 4 MgATP, and 0.02 Alexa-568-hydrazide- sodium, pH 7.3 (265 mOsm). The membrane voltage of the target cell was amplified (A-M Systems Model 2400), digitized at 10 kHz (PCI- 6221, National Instruments), and saved to a computer (WinEDR, University of Strathclyde). Voltage measurements have been corrected for a 13-mV junction potential. Stimulation light pulses for CsChrimson-expressing cells were generated by a green channel of an LED illumination system (pE-300, CoolLED) and passed through two long-pass filters that have a 610 nm cut-off wavelength (Edmund Optics).

**Statistics and reproducibility**. Experimental flies and genetic controls were tested at the same condition, and data are collected from at least two independent experiments. Each confocal micrograph was from 5 to 10 samples, all of which showed the same results. Statistical analysis was performed with GraphPad Prism Software version 7.00 (GraphPad Software). A summary of statistical analyses including adjusted p values is provided in Supplementary Table 1.

**Reporting summary**. Further information on research design is available in the Nature Research Reporting Summary linked to this article.

## Data availability
The source data underlying Supplementary Figs. 1b–g, 2h, 3a, b, 4a–d, 6a, 7a–h, and 8a, b are provided as a Source data file. Additional raw data comprising confocal micrograph stacks and electrophysiological recordings are available from https://doi.org/10.6084/m9.figshare.18737786. Source data are provided with this paper.

## Code availability
A MATLAB script used for FRIC analysis is available from https://github.com/daeyeonkim/FLIC.

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

## Acknowledgements

We thank I. Daubnerová (Slovak Academy of Sciences) for technical assistance. We thank B.J. Dickson (Janelia Research Campus), R. Niwa (University of Tsukuba), Y. Kwon (Korea Research Institute of Bioscience and Biotechnology), and D. Žitňan (Slovak Academy of Sciences) for reagents. This work was supported National Research Foundation of Korea (NRF) grants to Y.-J.K.; NRF-2015K2A1B8046794, NRF-2018R1A2A1A05079359, NRF-2019R1A4A1029724. This work was partially supported by the GIST Research Institute (GRI) grant funded by the GIST in 2022 (to Y.-J.K.), Institute of Information & communications Technology Planning & Evaluation (IITP) grant funded by the Korea government (MSIT) (No. 2020-0-01373, Artificial Intelligence Graduate School Program (Hanyang University)) (to A.J.K), and NIH grant R01AI04554 (to F.G.N). Stocks were obtained from the Korea Drosophila Resource Center (NRF-2017M3A9B8069650), the Bloomington Drosophila Stock Center and the Vienna Drosophila Resource Center.

## Author contributions

Conceived and designed the experiment: C.Z. and Y.-J.K; performed the experiments: C.Z. and A.J.K; analyzed the data: C.Z. and Y.-J.K.; contributed reagents and analysis: C.R.-P. and F.N.; wrote the manuscript: C.Z. and Y.-J.K.

## Competing interests

The authors declare no competing interests.
