## [Peer Review File · Nature Communications]

The insect somatostatin pathway gates vitellogenesis progression during reproductive maturation and the post-mating responseREVIEWER COMMENTS

Reviewer #1 (Remarks to the Author):

This is an excellent and extensive piece of work that outlines the role of the AstC-mTH neurons in the early stages of vitellogenesis. The ms is well-written and while it takes no prisoners in terms of the sophisticated *Drosophila* genetics that a general reader will not be able to follow (so some help is required in a general journal like NC) , it provides a significant advance in our knowledge of vitellogenesis in *Drosophila*. My only disappointment is in the naivety of the statistical analyses, which are in some places completely inappropriate (in fact incorrect). I suspect the correct analyses will give the same results (perhaps with one exception – see below). The Discussion is overlong and rambling and while it contains many interesting points, could be trimmed. Here are some specific comments

L118 Fig 1a. Is it really a two-stage process? Seems more like a gradual one in which early follicles turn to late but it takes a couple of days for the early-late transition to kick in –thereafter new early follicles keep being generated at ~10 per female and it does not come to an end as stated in the text l120-121 after 72 h. Are there any sign diffs between early follicles at 72- 120h?

Fig 1a, (indeed all figures) are the error bars representing SEM, SD, confidence limits.....?

In Fig 1a, the authors use one-way ANOVA separately at each time and on each set of stages. A better way to analyse the data would be to use a two-way ANOVA for each stage, with Genotypes and Time as the variables. This would use all the data for each stage and would generate interactions between Genotype and Time which could be followed up with post-hoc tests. For example if we take stage 8-11, a significant interaction would suggest that the differences in Genotypes we see at 12 and 24 h are significantly altered at later stages. I'm sure they would be but it's so much more appropriate that stating that the post-hoc tests for each individual ANOVA at different times 'look different'.

This also applies to other Figures such as Fig S1b, Fig 3e, Fig 5a where a two way ANOVA is more informative. This would not apply to Figs 4d,e,f in which the different stages are interdependent.

L131-2. There are significant effects in the overexpression of Ast-C at 72 and 96 h in Fig 1a so the authors are over-simplifying.

Fig 1C. I know it's obvious that Ast-C has an effect but the use of multiple unpaired t-tests are not the way to analyse such data. A two way ANOVA with genotype and TrpA as variables will generate an interaction (not using the UAS-TrpA1 control) – that is the correct way to analyse these data. Alternatively, use one-way ANOVA plus post hoc tests if you want to include the UAS control. I note that 30oC also gives a larger number of mature eggs ~50 than 25oC (~35 from Fig 1a) which is perhaps worth mentioning as vitellogenesis is temperature-sensitive in general.

L 162 why 'remarkably'?

L176 The authors do not show the expression patterns of Ast-C-D Gal4 but then 'restrict further' any pattern using a split gal4. It's not obvious to the reader why a split gal4 with a larger enhancer (Ast-C) would restrict expression further in concert with the smaller AstC-D construct? Help the general reader here.

L188-195 The rationale for using OtdFLP is not explained at all. From Fig 2a, total number of eggs in ovaries is repressed using TrpA1 with AstC-split but the subset of these AstC-D-split neurons that overlap with Otd neurons do not. Consequently 'Unlike the larger population of AstC-D-split-Gal4 neurons, OtdFLP-positive brain neurons had a limited impact on total oogenesis (Fig. 2d, Fig. S2b).

This suggests the AstC-D-split-Gal4 neurons in the brain are not important for this particular outcome.' Makes no sense.

Fig S3 –w1118 are not the correct controls for split-UASTrpA

L234 *Kr-h1* in italics

L261-2. Could there be another interneuron between AstC-mTh and the CA??

L280-287 - the effect on neuronal activity with SP and SP-less males is marginal and

needs to be analysed with a two-way ANOVA as the authors are wanting to state that b is significant but c is not. –the infamous cliff edge effect, ie you narrowly get significance with one comparison but not with another so you assume the two effects are different seems to be happening here. A significant interaction would support the authors' conclusions. Given the overlap in scores in Fig 4b, Fig 4a might be seen as representative or that the authors carefully selected the two images. Perhaps a caveat in the legend required here.

L368-369. Given that the SAG Ast-C mTh neuronal interactions, seems a bit disappointing to end on this note. Any ideas on the type of connection?

Discussion is way too long. Maybe a cartoon with a model would be helpful?

Otherwise, I really enjoyed reading this ms – it's a tour de force but needs to be made a little more accessible to the general readership

Reviewer #2 (Remarks to the Author):

Although the gonadotropic action of JH in insects has long been recognized, the mechanism whereby its timely synthesis and release are coordinated remains less well characterized. The present paper unraveled that a subset of neurons (termed AstC-mth) that synthesize Allatostatin-C (AstC) in the ventral nerve cord contribute to temporal coordination of ovarian maturation and mating via an inhibitory action of AstC on JH synthesis in corpora allata. The authors further put forward the hypothesis that the descending abdominal interneurons SAGs tonically activate AstC-mth neurons, which are deactivated upon mating as a result of diminished discharges of SAGs induced by the action of sex peptide (SP), a male seminal fluid component transferred to the female during copulation. While key roles of AstC-mth neurons in temporal control of JH synthesis were rigorously demonstrated, there was some ambiguity concerning the interplay between SAG-AstC-mth neurons.

Specific points

- 1. In Figures 4d and 4e, roles of AstC-mth neurons in vitellogenesis in virgin and mated females were tested by their artificial activation and conditional synaptic block, respectively. Whereas forced activation of AstC-mth neurons in mated females increased, as expected, the number of stage-10 oocytes, block of synaptic outputs from AstC-mth neurons in virgin females failed in increasing the number of stage-10 oocytes. This latter observation seems to be incompatible with the authors' model that Ast-C-mth neurons suppress vitellogenesis by downregulating JH synthetic activity. The authors need to suggest, at least, a possible explanation as to why stimulation induced retardation but synaptic block had no detectable effect.**
- 2. TRIC assays to monitor neural activities in AstC-mth detected only a marginal (yet significant at $P > 0.05$) difference between virgin and mated females in Figure 4b.**
- 3. In Figure 5c, simultaneous SAG hyperactivation and AstC-mth suppression was employed to determine "epistasis" between the two sets of neurons. Because the number of mature eggs retained at a high level even under such conditions, the authors placed AstC-mth neurons in an output path of SAG neurons. The design of this experiment may not be best suited for demonstrating the interplay between SAG and AstC-mth neurons, as the firing rate of SAG neurons in virgin females is likely high even without hyperactivation and thus dTrpA1-mediated stimulation could have only a limited effect. Why did not the authors attempt simultaneous SAG activation and AstC-mth activation instead? Also, it is preferable to have an additional control group of flies in which SAG-Gal4, UAS-dTrpA1 and SAG-LexA are all there but without UAS-Kir2.1.**
- 4. Trans-Tango assays did not yield any support for the direct synaptic contacts between SAG and AstC-mth neurons (Supplementary Figure 7). In view of the fact that SAGs are fast conducting descending neurons, conventional chemical synapses are most likely interface between the SAGs and their targets, and the failure in detecting trans-Tango signals would thus mean that SAGs and AstC-mTh are not direct synaptic partners for each other.**
- 5. Some experiments should include additional control groups.**

Minor points.

P1 line 34. "activation" must be "deactivation".

P4 line 11. Remove "initiation".

P7 line 1. Insert "in" between "arborizes" and "the".

P10 line 6. "functional" must be "functionally".

P15. Line 27. Insert "as" between "such" and "somatic".

Figure 3f. The bottom line of genotype descriptions should be numbered "7".

Reviewer #3 (Remarks to the Author):

In this manuscript, Zhang and co-workers, identify a new central neural circuit comprising Allatostatin-C producing neurons which regulate initiation of vitellogenesis in female ovaries post-eclosure and re-initiation after mating, via the regulation of the synthesis of juvenile hormone in the corpora allata.

I think this could be an extremely relevant study which substantially adds to our understanding of how central nervous system hormonally coordinates reproduction with internal states of the animal. There are however two main concerns that I think should be addressed before publication:

1) It is not clear to me how the dysregulation of this middle stage of oogenesis post-eclosure would impact on female's overall reproduction. As the authors refer in the discussion, intuitively it makes sense to have a mechanism at play which halts oogenesis until females fully reach reproductive maturation. This hypothesis and the relevance of this mechanism should be tested. I would guess that the premature vitellogenesis could result in the production of less viable eggs, so I would suggest the authors to devise some experiments where this would be addressed, such as measuring egg viability or progeny after eclosure.

2) One important aspect that is associated with the eclosure of the adult from the pupal case and the authors completely disregard is the re-initiation of feeding. Also, several genetic manipulations and different mating states strongly impact nutrient appetites. Moreover, feeding, particularly of protein, has a clear impact in oogenesis progression. Taken together, it will be critical to test if: 1) feeding initiation is upstream ActC cascade of events by checking how activity of these neurons is modulated by food availability post-eclosure, and 2) if resuming oogenesis in the first hours post-eclosure is dependent on food availability. If there is indeed a link between AstC regulation and food intake, than this mechanism would nicely coordinate oogenesis re-entry to nutrient availability. Authors should also be aware that the induction of the mating status by copulation with males or manipulation of SAG neurons, have both an impact on feeding, which could indirectly impact oogenesis and explain some of the observed results. I would thus suggest that food intake should be measured in key experiments to rule this out this possibility and put these findings in context to what has previously been published by multiple labs.

Minor comments/concerns:

- I felt some references were missing, especially in the introduction**
- Because all the genetic manipulations were performed throughout development it would be relevant to indicate if there are any developmental defects which could explain the phenotypes (e.g. Delay in eclosure times).**
- In general it will be important to have the expression of the different Gal4's and slits as supplemental data.**
- For some experiments it will important to better discriminate the observed effects on vitellogenesis vs egg maturation vs oviposition, in order to better understand and discuss the underlying neuronal mechanisms regulating oogenesis (see below some**

suggestions).

- Fig 1:

- Authors should show representative examples of ovarioles used for constructing the graph of AstC mutant females (at least) (a). It is also intriguing how AstC mutants increase the number of follicles during the first day after eclosure but this phenotype is not reflected in the number of mature oocytes after a couple of days. This means that either the flies are laying more eggs, or that they are undergoing apoptosis at some stage before full maturation. It will be important to include this information (as done for Fig 4 (OVI)).

- In experiments performed in c) and e), the phenotypes can be explained by an increase in oviposition or a decrease in oogenesis. It will be critical to indicate in these experiments how many eggs are actually being laid, as this is relevant for understanding the underlying mechanisms explaining the observed phenotype.

- Fig 2:

- Adding a short description of the genotype, driver used, in the different confocal panels would help the reader to interpret the figure.

- Fig 4:

- In d) I would expect, on the same note as my comment for Fig 1, that this decrease in the number of follicles will be reflected in the number of eggs laid. I guess this phenotype is not visible because in this experiment there was not enough time for the effect of the neuronal manipulation on follicle production to be reflected in the number of eggs laid. I think this should be tested.

- Fig 4 and Fig 5: Not clear why in some experiments the authors use SAG-Gal4 driver and in others they use SAG-split1-Gal4 and I could not find the difference between them within the text. This should be clarified.

Changes that should be considered in the abstract:

- "a pair of thoracic ganglion neurons that produce the neuropeptide allatostatin C (AstC-mTh)" – I would be careful with this statement. This is not shown. The evidence presented suggests that one pair of neurons show changes in activity as response to mating, but several neurons produce AstC, from the presented data using the antibody.

- "During sexual maturation, which takes place shortly after eclosion, AstC-mTh neurons are activated by excitatory inputs from SP abdominal ganglion (SAG) neurons" – I agree this is the most parsimonious explanation given the presented data but this was not formally tested. For this, TRIC could be performed on AstC neurons upon SAG neuron manipulations. I would rephrase this.

- "Upon mating, however, SP inhibits SAG neurons, leading to AstC-mTh neuronal activation." – My first comment is the same as for the previous point. Second, from what I understood from the authors' model and the data presented, mating should lead to SAG neurons' inhibition which in turn would lead to a decrease in AstC neuronal activity, not activation.

Point-by-point response

We would like to thank the Reviewers for their constructive comments. We have completed the new experiments they requested and revised the manuscript with new data. Below, we address each of the concerns raised by the referees in detail; our point-by-point responses are **blue**. The corresponding changes in the manuscript are **red**.

Reviewer #1 (Remarks to the Author):

This is an excellent and extensive piece of work that outlines the role of the AstC-mTH neurons in the early stages of vitellogenesis. The ms is well-written and while it takes no prisoners in terms of the sophisticated *Drosophila* genetics that a general reader will not be able to follow (so some help is required in a general journal like NC), it provides a significant advance in our knowledge of vitellogenesis in *Drosophila*. My only disappointment is in the naivety of the statistical analyses, which are in some places completely inappropriate (in fact incorrect). I suspect the correct analyses will give the same results (perhaps with one exception – see below).

The Discussion is overlong and rambling and while it contains many interesting points, could be trimmed. Here are some specific comments

We are grateful that our manuscript was well-received by Reviewer 1. We made the suggested changes to our statistical analyses and trimmed the Discussion. We also revised the figures and legends to improve accessibility for a general readership.

1. L118 Fig 1a. Is it really a two-stage process? Seems more like a gradual one in which early follicles turn to late but it takes a couple of days for the early-late transition to kick in –thereafter new early follicles keep being generated at ~10 per female and it does not come to an end as stated in the text 1120-121 after 72 h. Are there any sign diffs between early follicles at 72- 120h?

To illustrate our point more clearly, we re-plotted the control data of Fig. 1a in Fig. R1 below. This figure shows the number of both early and late vitellogenic follicles at the indicated number of hours after eclosion (HAE). Notably, the increase in the rate of production of vitellogenic follicles occurs in two phases. During the first phase between 0 and 48 HAE, the rate increased ~0.7 follicles per hour. This is higher than the rate of increase during the second phase between 48 and 120 HAE, which was ~0.2 follicles per hour. Furthermore, please also note that significantly fewer follicles enter early vitellogenesis in the second phase after 48 HAE.

Fig. 1R. Number of stage 8–11 (yellow) and 12–14 (orange) follicles per virgin female of the control strain (*AstC^{1/+}*, *AstC-Gal4/+*) at the indicated number of hours after eclosion (HAE). These are replotted from data shown in Fig. 1a.

Fig 1a, (indeed all figures) are the error bars representing SEM, SD, confidence limits.....?

Unless otherwise noted, the error bars represent SEM in all figures. The legends have been revised accordingly.

2. In Fig 1a, the authors use one-way ANOVA separately at each time and on each set of stages. A better way to analyze the data would be to use a two-way ANOVA for each stage, with Genotypes and Time as the variables. This would use all the data for each stage and would generate interactions between Genotype and Time which could be followed up with post-hoc tests. For example, if we take stage 8-11, a significant interaction would suggest that the differences in Genotypes we see at 12 and 24 h are significantly altered at later stages. I'm sure they would be but it's so much more appropriate that stating that the post-hoc tests for each individual ANOVA at different times 'look different'.

This also applies to other Figures such as Fig S1b, Fig 3e, Fig 5a where a two-way ANOVA is more informative. This would not apply to Figs 4d, e, f in which the different stages are interdependent.

Thank you for this important comment. As suggested, we have examined interactions between genotype and time for each stage using a two-way ANOVA followed by Bonferroni post-hoc tests in Fig.1a, Fig.1e, Fig. 3e, Fig. 5a, Fig. 5b, and Fig. S1f. The details of our use of two-way ANOVA statistics are described in Supplementary Table S1.

3. L131-2. There are significant effects in the overexpression of *Ast-C* at 72 and 96 h in Fig 1a so the authors are over-simplifying.

The new statistical analysis (a two-way ANOVA followed by Bonferroni post-hoc tests) confirmed that *AstC* rescue (i.e., *AstC* overexpression) and the control groups do not produce significantly different numbers of early- and late-stage vitellogenic oocytes at any of the examined time points, including 72 and 96 HAE.

4. Fig 1C. I know it's obvious that Ast-C has an effect but the use of multiple unpaired t-tests are not the way to analyze such data. A two-way ANOVA with genotype and TrpA as variables will generate an interaction (not using the UAS-TrpA1 control) – that is the correct way to analyse these data. Alternatively, use one-way ANOVA plus post hoc tests if you want to include the UAS control.

As suggested, we re-analyzed the data in Fig. 1c with a one-way ANOVA followed by Tukey's test for multiple comparisons.

I note that 30°C also gives a larger number of mature eggs ~50 than 25°C (~35 from Fig 1a) which is perhaps worth mentioning as vitellogenesis is temperature-sensitive in general.

Please note that Fig. 1a and Fig. 1c are from different cohorts of experiments and examined different genotypes. To get better insight into the temperature sensitivity of vitellogenesis, we examined the results of more carefully controlled experiments (see Fig. R2 below). We found that the 30°C incubation that activates dTrpA1 in AstC-D neurons suppressed vitellogenesis and reduced oogenesis in the experimental group (genotype 1), but not in the control genotypes (genotypes 2 and 3). There is a weak trend toward increased oocyte production in the control flies maintained at 30°C, but this effect is not strong enough to warrant a mention of temperature sensitivity of vitellogenesis in the manuscript.

Fig 2R. Vitellogenesis does not seem sensitive to ambient temperatures. Above, experimental protocol. Below, number of mature eggs per virgin female of the indicated genotype maintained for 3 days after eclosion at the indicated temperatures ($n = 40-60$). The letters above the bars indicate significant differences ($p < 0.05$) in comparisons of temperature and genotype (two-way ANOVA followed by Bonferroni post-hoc tests).

5. L 162 why 'remarkably'?

We removed it.

6. L176 The authors do not show the expression patterns of *AstC-D Gal4* but then 'restrict further' any pattern using a split *gal4*. It's not obvious to the reader why a split *gal4* with a larger enhancer (*AstC*) would restrict expression further in concert with the smaller *AstC-D* construct? Help the general reader here.

We agree with this suggestion to document the difference between *AstC-D-Gal4* and *AstC-D-split-Gal4* (now referred to as *AstC-mTh-Gal4*). In the newly prepared Figure (i.e., Fig S2a-c), we compared the expression patterns of *AstC-Gal4*, *AstC-D-Gal4*, and *AstC-mTh-Gal4* side-by-side. It is now clear that the split-Gal4 significantly restricted *AstC-D-Gal4* expression. We added a brief description in lines 196-199.

7. L188-195 The rationale for using *OtdFLP* is not explained at all. From Fig 2a, total number of eggs in ovaries is repressed using *TrpA1* with *AstC-split* but the subset of these *AstC-D-split* neurons that overlap with *Otd* neurons do not. Consequently 'Unlike the larger population of *AstC-D-split-Gal4* neurons, *OtdFLP*-positive brain neurons had a limited impact on total oogenesis (Fig. 2d, Fig. S2b). This suggests the *AstC-D-split-Gal4* neurons in the brain are not important for this particular outcome.' Makes no sense.

In response to this comment, we revised lines 211-216 as follows:

"To visualize and examine the function of *AstC-mTh-Gal4* neurons in the brain, we combined *AstC-mTh-Gal4* with the *OtdFLP* transgene^{23,24}, which is active exclusively in the brain, not the VNC (Fig. S2f). Unlike activation of *AstC-mTh-Gal4* neurons (including those in the VNC), activation of *AstC-mTh-Gal4* and *OtdFLP* double-positive brain neurons had a limited impact on total oogenesis (Fig. 2d). This suggests the *AstC-mTh-Gal4* neurons in the brain are unlikely responsible for oogenesis inhibition."

8. Fig S3 –*w1118* are not the correct controls for split-*UASTrpA*

Due to the COVID-19 knockdown, we were unable to run additional biochemical analyses quantifying JH with additional controls. Thus, we examined *Kh-h1* expression in all the necessary genotypes (*w¹¹¹⁸*, a control, and a test group) and compared it side-by-side with currently available JH quantification data (Fig S3A and 3B). Our new experiments confirmed that *w¹¹¹⁸* and a control (*UAS-dTrpA1/+*) produced very similar levels of *Kh-h1* expression. This supports our argument that *w¹¹¹⁸* in Fig. S3A and the correct controls (*UAS-dTrpA1/+* and *AstC-mTh-Gal4/+*) have similar levels of JH. We made this clear in the legend for Fig. S3.

9. L234 *Kr-h1* in italics

We corrected this mistake.

10. L261-2. Could there be another (*AstC-positive*) interneuron between *AstC-mTh* and the CA??

This is unlikely. The expression of *AstC receptor-RNAi* in the CA produced a phenotype comparable to that of *AstC* mutants. Thus, any potential relay neurons connecting the *AstC-mTh* neurons and the CA should also be *AstC-positive*. We did not observe any *AstC-positive* processes innervating the CA (shown in the new Fig. S4b). We have revised the manuscript accordingly at lines 303-304.

11. L280-287 - the effect on neuronal activity with SP and SP-less males is marginal and needs to be analysed with a two-way ANOVA as the authors are wanting to state that b is significant but c is not. –the infamous cliff edge effect, if you narrowly get significance with one comparison but not with another so you assume the two effects are different seems to be happening here. A significant interaction would support the authors' conclusions. Given the overlap in scores in Fig 4b, Fig 4a might be seen as representative or that the authors carefully selected the two images. Perhaps a caveat in the legend required here.

In our original submission, we examined mating and SP effects in two separate cohorts of experiments. To avoid the potential caveat pointed out in this comment, we performed new experiments examining both effects in a single cohort of experiments. In the new experiment, we also measured nucleus-targeted RFP activity together with TRIC. This allowed us to control for individual variations in GAL4 activity by normalizing the TRIC signal with the ncRFP signal. Our new results, which now appear in the revised Fig. 4a, convincingly demonstrate that mating suppresses intracellular Ca^{2+} activity in *AstC-mTh* via SP. We have updated the Materials and Methods section accordingly (see lines 666-670).

12. L368-369. Given that the SAG *AstC-mTh* neuronal interactions, seems a bit disappointing to end on this note. Any ideas on the type of connection?

In this revision, we used electrophysiological patch recordings of *AstC-mTh* neurons and asked whether a minute-long optogenetic activation of SAG via the channelrhodopsin variant csChrimson triggered action potentials in *AstC-mTh* neurons. In this new experiment, we found that short-term activation of SAG neurons did not depolarize *AstC-mTh* neurons in the mated females (see lines 429-441). Our preliminary analysis also found the same in the virgin females (not shown). Thus, we propose that there are additional unknown neuron(s) that connect SAG and *AstC-mTh* neurons, which seem to require long-term excitatory inputs from SAG (hours instead of minutes) to relay a signal to the *AstC-mTh* neurons.

13. Discussion is way too long. Maybe a cartoon with a model would be helpful?

We have trimmed the Discussion section significantly. It now has ~25% fewer words than the previous version (1,492 vs. 1,971 words). We also added two summary cartoons to explain our models in the new Figs. 4e and 5f.

Otherwise, I really enjoyed reading this ms – it's a tour de force but needs to be made a little more accessible to the general readership

To improve our manuscript's accessibility for a general readership, we simplified the genotype descriptions in the Figures and moved the full descriptions of the genotypes to supplemental Table S2.

Reviewer #2 (Remarks to the Author):

Although the gonadotropic action of JH in insects has long been recognized, the mechanism whereby its timely synthesis and release are coordinated remains less well characterized. The present paper unraveled that a subset of neurons (termed *AstC-mth*) that synthesize Allatostatin-C (*AstC*) in the ventral nerve cord contribute to temporal coordination of ovarian maturation and mating via an inhibitory action of *AstC* on JH synthesis in corpora allata. The authors further put forward the hypothesis that the descending abdominal interneurons SAGs tonically activate *AstC-mth* neurons, which are deactivated

upon mating as a result of diminished discharges of SAGs induced by the action of sex peptide (SP), a male seminal fluid component transferred to the female during copulation. While key roles of AstC-mth neurons in temporal control of JH synthesis were rigorously demonstrated, there was some ambiguity concerning the interplay between SAG-AstC-mth neurons.

We are grateful that our work is well-received by Reviewer 2. In this revision, we have refined our understanding of the functional relationship between SAG neurons and *AstC-mTh* neurons in virgin and mated females (see our responses to comments #1, #3, and #4) with new experiments (Figs. 4c and 5d). In addition, we have included electrophysiological patch recordings of *AstC-mTh* neurons and more physiological evidence indicating that it is unlikely that SAG and *AstC-mTh* neurons form direct synaptic connections.

Specific points

1. *In Figures 4d and 4e, roles of AstC-mth neurons in vitellogenesis in virgin and mated females were tested by their artificial activation and conditional synaptic block, respectively. Whereas forced activation of AstC-mth neurons in mated females increased, as expected, the number of stage-10 oocytes, block of synaptic outputs from AstC-mth neurons in virgin females failed in increasing the number of stage-10 oocytes. This latter observation seems to be incompatible with the authors' model that AstC-mth neurons suppress vitellogenesis by downregulating JH synthetic activity. The authors need to suggest, at least, a possible explanation as to why stimulation induced retardation but synaptic block had no detectable effect.*

The forced activation of SAG neurons or *AstC-mTh* neurons in mated females led to a significant ~50% reduction in post-mating vitellogenesis and in the number of stage 10 oocytes. With this result and others, we propose the model that follows. In mature virgin females, *AstC-mTh* neurons continue to supply inhibitory inputs to the CA, suppressing its production of JH. In mated females, SP reduces the neural activity of *AstC-mTh* neurons by silencing SAG neurons. Thus, the reduced activity of *AstC-mTh* neurons in mated females disinhibits CA activity, thereby permitting its production of JH and subsequent post-mating vitellogenesis. As noted by the reviewer, however, disinhibition alone (i.e., silencing *AstC-mTh* neurons) in virgin females did not stimulate vitellogenesis. To explain this, we proposed that the mating signal (i.e., SP) both stimulates the CA while also relieving inhibition of it (i.e., from AstC). If the CA receives no stimulatory input (i.e., in mature virgin females), removal of the inhibitory input alone (i.e., silencing *AstC-mTh* neurons) does not stimulate the CA and JH production.

In this revision of the manuscript, we sought to verify this model by identifying the stimulatory input to the CA. To our knowledge, ecdysis triggering hormone (ETH) is the only stimulatory factor for the CA in fruit flies (Meiselman et al., 2017). SP induces 20E biosynthesis via the neuronal SP response pathway (Ameku et al., 2016), and 20E activates ETH expression and secretion from adult Inka cells (Meiselman et al., 2017). Thus, we asked whether ETH signaling plays a role in activating post-mating vitellogenesis by knocking-down the ETH receptor (ETHR) in the CA. As expected, *ETHR-RNAi* in the CA reduced the number of stage 10 oocytes by ~50% (newly prepared Fig. 4c). This observation is consistent with the model described above. In the newly prepared Fig. 4e, we expanded our model in which SP stimulates post-mating vitellogenesis by simultaneously enhancing ETH-induced stimulation of the CA and reducing AstC-induced inhibition of the CA (Fig. 4e). We have now included a description of these results on lines 358-366.

2. *TRIC assays to monitor neural activities in AstC-mth detected only a marginal (yet significant at $P>0.05$) difference between virgin and mated females in Figure 4b.*

In our original submission, we examined mating and SP effects in two separate cohorts of experiments. To avoid the potential caveat pointed out in this comment, we performed new experiments examining both effects in a single cohort of experiments. In the new experiment, we also measured nucleus-targeted RFP activity together with TRIC. This allowed us to control for individual variations in GAL4 activity by normalizing the TRIC signal with the ncRFP signal. Our new results, which now appear in the revised Fig. 4a, convincingly demonstrate that mating suppresses intracellular Ca^{2+} activity in *AstC-mTh* via SP. We have updated the Materials and Methods section accordingly (see lines 666-670).

3. In Figure 5c, simultaneous SAG hyperactivation and AstC-mth suppression was employed to determine “epistasis” between the two sets of neurons. Because the number of mature eggs retained at a high level even under such conditions, the authors placed AstC-mth neurons in an output path of SAG neurons. The design of this experiment may not be best suited for demonstrating the interplay between SAG and AstC-mth neurons, as the firing rate of SAG neurons in virgin females is likely high even without hyperactivation and thus dTrpA1-mediated stimulation could have only a limited effect. Why did not the authors attempt simultaneous SAG activation and AstC-mth activation instead? Also, it is preferable to have an additional control group of flies in which SAG-Gal4, UAS-dTrpA1 and SAG-LexA are all there but without UAS-Kir2.1.

Because the SAG neurons of virgin females are active even without forced activation, the reviewer suggested we design new experiments examining the functional relationship between SAG and *AstC-mTh* neurons (original Fig. 5c).

In mated females, we found that thermal activation of SAG neurons or *AstC-mTh* neurons produced almost the same phenotype (compare Fig. 4b and 4d). Thus, we wanted to show that the effect caused by manipulation of the SAG neurons could be blocked by the opposite manipulation of *AstC-mTh* neurons. As Reviewer 2 points out, the firing rate of the SAG neurons in virgin females is presumably high. Thus, we opted to silence the SAG neurons of virgin females. We found that this increased the number of early vitellogenic oocytes during the first day after eclosion (Fig. 5a, b). We then simultaneously silenced SAG neurons and activated *AstC-mTh* neurons in these females to determine whether activation of *AstC-mTh* neurons overrides the vitellogenesis-stimulating effect of SAG silencing (Fig. 5d). Indeed, we found activation of *AstC-mTh* neurons reduced the increased number of early vitellogenic oocytes to control levels, consistent with the hypothesis that the *AstC-mTh* neurons function downstream of the SAG neurons (see lines 402-406).

This observation is further complemented by another experiment with opposite activity manipulations, the simultaneous activation of the SAG neurons and inhibition of the *AstC-mTh* neurons (Fig. 5c). Long-term activation of SAG neurons reduced total oogenesis output during reproductive maturation (presumably due to reduced vitellogenesis), but simultaneous silencing of *AstC-mTh* neurons blocked this SAG effect and restored oogenesis output to control levels. The data we presented in our first submission lacked a few control genotypes. In this revision, we have performed new experiments and added some important controls (newly prepared Fig. 5c).

4. Trans-Tango assays did not yield any support for the direct synaptic contacts between SAG and AstC-mth neurons (Supplementary Figure 8). In view of the fact that SAGs are fast conducting descending neurons, conventional chemical synapses are most likely interface between the SAGs and their targets, and the failure in detecting trans-Tango signals would thus mean that SAGs and AstC-mTh are not direct synaptic partners for each other.

In this revision, we performed electrophysiological patch recordings of *AstC-mTh* neurons. We asked whether a minute-long optogenetic activation of SAG via the channelrodopsin variant csChrimson could trigger action potentials in *AstC-mTh* neurons. In this new experiment, we found that short-term activation of SAG neurons did not depolarize *AstC-mTh* neurons in the mated females (see lines 429-441). Our

preliminary analysis also found the same in the virgin females (not shown). With this result and our Trans-Tango results, we have concluded that additional unknown neuron(s) connect SAG and *AstC-mTh*. These unknown neurons seem to require long-term excitatory inputs from SAG (hours instead of minutes) to relay a signal to *AstC-mTh* neurons.

5. Some experiments should include additional control groups.

We performed new experiments and included additional control groups in Figs. 3a, 4a, 5c, and S3b.

Minor points.

P1 line 34. "activation" must be "deactivation".

We have corrected this mistake.

P4 line 11. Remove "initiation".

The number of early vitellogenic follicles drops at 72 HAE, whereas the number of late vitellogenic follicles continue to increase gradually to 120 HAE (Fig 1a). This is why we would rather say "vitellogenesis initiation slows down until coming to an end 72 hours after eclosion."

P7 line 1. Insert "in" between "arborizes" and "the".

Revised as suggested.

P10 line 6. "functional" must be "functionally".

Revised as suggested.

P15. Line 27. Insert "as" between "such" and "somatic".

Revised as suggested.

Figure 3f. The bottom line of genotype descriptions should be numbered "7".

Revised. We now use a color code for the genotypes in the bar graph (see Fig. 3f).

Reviewer #3 (Remarks to the Author):

In this manuscript, Zhang and co-workers, identify a new central neural circuit comprising Allatostatin-C producing neurons which regulate initiation of vitellogenesis in female ovaries post-eclosure and re-initiation after mating, via the regulation of the synthesis of juvenile hormone in the corpora allata.

I think this could be an extremely relevant study which substantially adds to our understanding of how central nervous system hormonally coordinates reproduction with internal states of the animal. There are however two main concerns that I think should be addressed before publication:

1) *It is not clear to me how the dysregulation of this middle stage of oogenesis post-eclosure would impact on female's overall reproduction. As the authors refer in the discussion, intuitively it makes sense to have a mechanism at play which halts oogenesis until females fully reach reproductive maturation. This hypothesis and the relevance of this mechanism should be tested.*

I would guess that the premature vitellogenesis could result in the production of less viable eggs, so I would suggest the authors to devise some experiments where this would be addressed, such as measuring egg viability or progeny after eclosure.

To address this concern, we performed new experiments examining egg-to-pupa and pupa-to-adult development rates in AstC-deficient mutant and control groups (Fig. S1c). Our results suggest the precocious vitellogenesis caused by AstC deficiency does not seem to compromise the viability of eggs or progeny. See lines 136-138.

2) *One important aspect that is associated with the eclosure of the adult from the pupal case and the authors completely disregard is the re-initiation of feeding. Also, several genetic manipulations and different mating states strongly impact nutrient appetites. Moreover, feeding, particularly of protein, has a clear impact in oogenesis progression. Taken together, it will be critical to test:*

1. *If feeding initiation is upstream ActC cascade of events by checking how activity of these neurons is modulated by food availability post-eclosure,*
2. *If resuming oogenesis in the first hours post-eclosure is dependent on food availability. If there is indeed a link between AstC regulation and food intake, then this mechanism would nicely coordinate oogenesis re-entry to nutrient availability.*

Authors should also be aware that the induction of the mating status by copulation with males or manipulation of SAG neurons, have both an impact on feeding, which could indirectly impact oogenesis and explain some of the observed results. I would thus suggest that food intake should be measured in key experiments to rule this out this possibility and put these findings in context to what has previously been published by multiple labs.

We appreciate this comment addressing the potential link between feeding and vitellogenesis. During our revision process, we became aware of a preprint manuscript describing the function of AstC in the starvation-induced feeding of females (Kubrak et al., 2021). Briefly, AstC from gut enteroendocrine cells promotes food-seeking behavior in starved females. Although this study found little role for CNS-derived AstC, we still decided to perform a series of experiments to determine whether AstC deficiency promotes vitellogenesis indirectly by enhancing feeding. In our new experiments, however, we found both AstC-deficient mutant and control groups engage in low levels of feeding activity during reproductive maturation. Both groups showed no significant difference in any of the feeding parameters measured in the FLIC assay, an automated feeding monitoring assay (Fig. S4a). In mature females (i.e., 4–5 DAE), we noted that AstC deficiency induced a moderate increase in feeding activity. *AstC-mTh* neuron silencing in mature females, however, had almost no impact on feeding (Fig. S4b). As suggested by Reviewer 3, we asked whether *AstC-mTh* neural activity changes in response to feeding conditions during reproductive maturation and found that it does not (Fig. S4c). Thus, we have concluded that the vitellogenesis phenotype of AstC-deficient females is not secondary to changes in feeding. The new experiments mentioned above are now described on lines 281–295 of the revised manuscript.

Minor comments/concerns:

- *I felt some references were missing, especially in the introduction*

We inserted ref #2, #3, #11-14 in the revised introduction.

- *Because all the genetic manipulations were performed throughout development it would be relevant to indicate if there are any developmental defects which could explain the phenotypes (e.g. Delay in eclosure times).*

Note that most of our genetic experiments were complemented by temporal activation or silencing experiments (i.e., dTrap1 or Shi^{ts} experiments; Figs. 1c, 1d, 2a, 2d, 3a, 3b, 4b, 4d, 5b, 5c, S1g, S8b, etc.), which typically show very limited effects on development. We did, however, carefully examine development in the *AstC* mutant and *AstC-mTh* neuron-silenced flies and observed no measurable developmental abnormalities (not shown).

- *In general, it will be important to have the expression of the different Gal4's and splits as supplemental data.*

We agree that it is important to document the difference between *AstC-D-Gal4* and *AstC-D-split-Gal4* (now refer to as *AstC-mTh-Gal4*). In the newly prepared Figures S2a-c, we compare the expression patterns of *AstC-Gal4*, *AstC-D-Gal4*, and *AstC-mTh-Gal4* side-by-side. It is now clearer that the split-Gal4 significantly restricts *AstC-D-Gal4* expression. We have added a short description of these results to lines 196–199.

- *For some experiments it will important to better discriminate the observed effects on vitellogenesis vs egg maturation vs oviposition, in order to better understand and discuss the underlying neuronal mechanisms regulating oogenesis (see below some suggestions).*

- Fig 1:

o Authors should show representative examples of ovarioles used for constructing the graph of AstC mutant females (at least) (a).

We now include confocal images of ovarioles from *AstC* mutant, control and rescue females in the revised manuscript (Fig. S1a).

It is also intriguing how AstC mutants increase the number of follicles during the first day after eclosure but this phenotype is not reflected in the number of mature oocytes after a couple of days. This means that either the flies are laying more eggs, or that they are undergoing apoptosis at some stage before full maturation. It will be important to include this information (as done for Fig 4 (OVI)).

Each ovary holds a limited number of mature oocytes; it houses 15–20 ovarioles with each ovariole containing 1–2 mature oocytes after the completion of reproductive maturation in *Drosophila*. When the ovary reaches capacity, it should stop producing vitellogenic follicles, regardless of genotype. This is what we observed in Fig. 1a. The ovaries of *AstC* mutant females seem to reach capacity earlier than controls, simply because they begin producing vitellogenic follicles earlier. This interpretation predicts that *AstC* mutant virgin females will not exhibit precocious egg laying. Indeed, when we examined virgin egg laying activity at several time points after eclosion matching those of Fig. 1a, we observed no measurable difference between *AstC* mutant and control females (see Fig. S1b). Both mutant and control virgins laid

almost no eggs during the first 72 hours after eclosion. Consistent with our hypothesis, *AstC-mTh* neuron silencing did not induce egg laying in virgin females (Fig. S1f). Thus, we concluded that the precocious vitellogenesis caused by *AstC*-deficiency does not cause egg laying during or after reproductive maturation.

o In experiments performed in c) and e), the phenotypes can be explained by an increase in oviposition or a decrease in oogenesis. It will be critical to indicate in these experiments how many eggs are actually being laid, as this is relevant for understanding the underlying mechanisms explaining the observed phenotype.

As suggested, we examined egg laying in virgin females with activated *AstC-mTh* neurons (i.e., *AstC-D>dTrpA1*, 30°C). We found activation of *AstC-mTh* neurons does not increase egg laying (Fig. S1e). Thus, it is unlikely that the reduction in mature eggs observed in *AstC-D>dTrpA1* females (30°C) is due to an increase in virgin egg laying activity.

• Fig 2:

o Adding a short description of the genotype, driver used, in the different confocal panels would help the reader to interpret the figure.

As suggested, we have added an indication of the genotype to each panel of the confocal images (Figs. 2, 3, S2, S5, and S9).

• Fig 4:

o In d) I would expect, on the same note as my comment for Fig 1, that this decrease in the number of follicles will be reflected in the number of eggs laid. I guess this phenotype is not visible because in this experiment there was not enough time for the effect of the neuronal manipulation on follicle production to be reflected in the number of eggs laid. I think this should be tested.

Reviewer 3 is asking us to test the hypothesis that increased oviposition caused by longer-term (i.e., 48 hours rather than 24 hours) activation of *AstC-mTh* neurons would facilitate vitellogenesis progression, resulting in fewer stage 10 follicles. As suggested, we performed a new experiment asking whether 48 hours of *AstC-mTh* neuron activation after mating increases oviposition. As we observed with the 24-hour activation, the 48-hour activation also reduced oviposition by 24% (see Ovi in Fig. S8a). This is consistent with our interpretation that activation of *AstC-mTh* reduces post-mating vitellogenesis and the number of mature eggs available for oviposition.

• Fig 4 and Fig 5: *Not clear why in some experiments the authors use SAG-Gal4 driver and in others they use SAG-split1-Gal4 and I could not find the difference between them within the text. This should be clarified.*

We clarified the differences in the legend of Fig. 5 (lines 1,205-1,211). Please note that we refer to SAG-split1-Gal4 as SAG-1-Gal4 in deference to the naming system used in the study that first reported this reagent.

Changes that should be considered in the abstract:

• *"a pair of thoracic ganglion neurons that produce the neuropeptide allatostatin C (*AstC-mTh*)" – I would be careful with this statement. This is not shown. The evidence presented suggests that one pair of*

neurons show changes in activity as response to mating, but several neurons produce AstC, from the presented data using the antibody.

In response to this comment, we replaced “a pair of thoracic ganglion neurons” with “a small subset of thoracic ganglion neurons.”

• "During sexual maturation, which takes place shortly after eclosion, AstC-mTh neurons are activated by excitatory inputs from SP abdominal ganglion (SAG) neurons" – I agree this is the most parsimonious explanation given the presented data but this was not formally tested. For this, TRIC could be performed on AstC neurons upon SAG neuron manipulations. I would rephrase this.

In response to this comment, we revised the sentence as follows:

“AstC-mTh neurons seem to receive excitatory inputs from SP abdominal ganglion (SAG) neurons.”

• "Upon mating, however, SP inhibits SAG neurons, leading to AstC-mTh neuronal activation." – My first comment is the same as for the previous point. Second, from what I understood from the authors' model and the data presented, mating should lead to SAG neurons' inhibition which in turn would lead to a decrease in AstC neuronal activity, not activation.

We have corrected this mistake. It was supposed to be silencing rather than activation.

REFERENCES

Meiselman, M. *et al.* Endocrine network essential for reproductive success in *Drosophila melanogaster*. *Proc. Natl. Acad. Sci. U. S. A.* **114**, E3849–E3858 (2017).

Ameku, T. & Niwa, R. Mating-induced increase in germline stem cells via the neuroendocrine system in female *Drosophila*. *PLoS Genet.* **12**, (2016).

Kubrak, O. *et al.* The gut hormone Allatostatin C regulates food intake and metabolic homeostasis under nutrient stress. Preprint at *bioRxiv* 2020.12.05.412874 (2020).

REVIEWERS' COMMENTS

Reviewer #1 (Remarks to the Author):

I am pleased that the authors have responded positively to my comments. The ms is much tighter and the statistical analyses much more elegant and robust. The extra experiments are informative and the analyses appropriate. This is an excellent piece of work and I congratulate the authors on performing such an interesting and extensive study. I have one minor comment

l165-6 'Neither AstC-D-Gal4neuronal activation nor silencing did not stimulate egg-laying in virgin females (Fig. S1e, f). This is awkward with 3 negatives, neither, nor, not. Suggest 'Neither AstC-D-Gal4neuronal activation nor silencing stimulated egg-laying in virgin females (Fig. S1e, f).

Reviewer #2 (Remarks to the Author):

All issues I raised have been adequately addressed. I have no further comments.

Reviewer #3 (Remarks to the Author):

Zhang and co-workers have now provided a series of new experiments and analyses which in the opinion of this reviewer improved the quality of the manuscript. Regarding my main concerns:

1) The authors have now provided new data where they show that egg viability is not affected by the precocious vitellogenesis induced by AstC deficiency, indicating that this mechanism does not impact the progeny outcomes of females. I think the authors address this point appropriately in the discussion and that this mechanism in the context of natural environment might lead to different outcomes than in laboratorial contexts.

2) My second main concern referred to the possibility of AstC impact on vitellogenesis being associated to changes occurring in feeding behavior during the post-eclosure and mating periods. In this revision process authors have tried to address this reviewer concern by providing new data on analysis of the feeding behavior of AstC mutant females and females where thoracic AstC neurons were activated as well as evaluating TRIC activity in these neurons in different nutritional states. Authors have used a highly quantitative method to measure feeding, unfortunately the sample size seems too low (I am assuming that each circle corresponds to individual flies because this information is not indicated either in the legend or in the methods section). I also believe it would have been more appropriate to use a protein food source rather than sucrose in this particular experiment. However, I do appreciate the effort of the authors in providing these data. I think these new data clearly point out a function for AstC in food intake, and a possible function of the thoracic AstC neurons – I know that significance was not reached in this experiment but with such low number of individuals tested I am not surprised. Unfortunately authors failed to provide data regarding the requirement of feeding initiation for vitellogenesis post-eclosure, or further data for some key manipulations including for SAG neurons. Having this said, this seems to be a complex interaction and I would not expect the authors to address it fully during the revision process.

Regarding my minor concerns and changes to the abstract, authors provided new data to address most of them including regarding the oviposition of multiple manipulations which further supports their claims, and did the appropriate changes to the text.

In line 116: "...post-eclosion (white bar...)", should be read yellow bar instead?

Overall, my opinion is that the findings provided by this manuscript are of general interest to the community providing novel understanding of how central nervous system hormonally coordinates

reproduction, and I would recommend it for publication.

Point-by-point response

We would like to thank all Reviewers for their positive comments. Followings are corrections we made in response to reviewers comments.

Reviewer #1 (Remarks to the Author):

I am pleased that the authors have responded positively to my comments. The ms is much tighter and the statistical analyses much more elegant and robust. The extra experiments are informative and the analyses appropriate. This is an excellent piece of work and I congratulate the authors on performing such an interesting and extensive study. I have one minor comment

l165-6 'Neither AstC-D-Gal4 neuronal activation nor silencing did not stimulate egg-laying in virgin females (Fig. S1e, f). This is awkward with 3 negatives, neither, nor, not. Suggest 'Neither AstC-D-Gal4 neuronal activation nor silencing stimulated egg-laying in virgin females (Fig. S1e, f).

We are grateful that our revision was satisfactory.

We revised lines 204-205, as suggested.

Reviewer #2 (Remarks to the Author):

All issues I raised have been adequately addressed. I have no further comments.

We are grateful that our revision was satisfactory

Reviewer #3 (Remarks to the Author):

Zhang and co-workers have now provided a series of new experiments and analyses which in the opinion of this reviewer improved the quality of the manuscript. Regarding my main concerns:

1) The authors have now provided new data where they show that egg viability is not affected by the precocious vitellogenesis induced by AstC deficiency, indicating that this mechanism does not impact the progeny outcomes of females. I think the authors address this point appropriately in the discussion and that this mechanism in the context of natural environment might lead to different outcomes than in laboratorial contexts.

2) My second main concern referred to the possibility of AstC impact on vitellogenesis being associated to changes occurring in feeding behavior during the post-eclosure and mating periods. In this revision process authors have tried to address this reviewer concern by providing new data on analysis of the feeding behavior of AstC mutant females and females where thoracic AstC neurons were activated as well as evaluating TRIC activity in these neurons in different nutritional states. Authors have used a highly quantitative method to measure feeding, unfortunately the sample size seems too low (I am assuming that each circle corresponds to individual flies because this information is not indicated either in the legend or in the methods section). I also believe it would have been more appropriate to use a protein

food source rather than sucrose in this particular experiment. However, I do appreciate the effort of the authors in providing these data. I think these new data clearly point out a function for AstC in food intake, and a possible function of the thoracic AstC neurons – I know that significance was not reached in this experiment but with such low number of individuals tested I am not surprised. Unfortunately, authors failed to provide data regarding the requirement of feeding initiation for vitellogenesis post-eclosure, or further data for some key manipulations including for SAG neurons. Having this said, this seems to be a complex interaction and I would not expect the authors to address it fully during the revision process.

Regarding my minor concerns and changes to the abstract, authors provided new data to address most of them including regarding the oviposition of multiple manipulations which further supports their claims, and did the appropriate changes to the text.

In line 116: "...post-eclosion (white bar...)", should be read yellow bar instead?

Overall, my opinion is that the findings provided by this manuscript are of general interest to the community providing novel understanding of how central nervous system hormonally coordinates reproduction, and I would recommend it for publication.

We are grateful that our revision was found satisfactory to the reviewer.

For the reviewer's second comment, we agree that future work is needed to address the precise function of AstC and *AstC-mTh* neurons in the context of feeding. However, we want to leave a note that both our study and others (i.e., Kubrak et al., 2020) failed to establish a causal relationship between neuronal AstC and feeding.

In response to the reviewers' other comments, we revised as follows.

1. In the legend of Figure S4, we included 'Each circle indicates individual flies examined'.
2. We corrected line 154 and 'white bar' now read as 'yellow circle'.

Reference

Kubrak, O. *et al.* The gut hormone Allatostatin C regulates food intake and metabolic homeostasis under nutrient stress. Preprint at *bioRxiv* 2020.12.05.412874 (2020).